# An extrasynaptic GABAergic signal modulates a pattern of forward movement in *Caenorhabditis elegans*

Yu Shen[1†], Quan Wen[2,3,4†], He Liu[1†], Connie Zhong[1], Yuqi Qin[1], Gareth Harris[1], Taizo Kawano[5‡], Min Wu[2], Tianqi Xu[2], Aravinthan DT Samuel[3*], Yun Zhang[1*]

[1]Department of Organismic and Evolutionary Biology, Center for Brain Science, Harvard University, Cambridge, United States; [2]Chinese Academy of Sciences Key Laboratory of Brain Function and Disease, School of Life Sciences, University of Science and Technology of China, Hefei, China; [3]Department of Physics, Center for Brain Science, Harvard University, Cambridge, United States; [4]CAS Center for Excellence in Brain Science and Intelligence Technology, University of Science and Technology of China, Hefei, China; [5]Lunenfeld-Tanenbaum Research Institute, Mount Sinai Hospital, University of Toronto, Toronto, Canada

**Abstract** As a common neurotransmitter in the nervous system, γ-aminobutyric acid (GABA) modulates locomotory patterns in both vertebrates and invertebrates. However, the signaling mechanisms underlying the behavioral effects of GABAergic modulation are not completely understood. Here, we demonstrate that a GABAergic signal in *C. elegans* modulates the amplitude of undulatory head bending through extrasynaptic neurotransmission and conserved metabotropic receptors. We show that the GABAergic RME head motor neurons generate undulatory activity patterns that correlate with head bending and the activity of RME causally links with head bending amplitude. The undulatory activity of RME is regulated by a pair of cholinergic head motor neurons SMD, which facilitate head bending, and inhibits SMD to limit head bending. The extrasynaptic neurotransmission between SMD and RME provides a gain control system to set head bending amplitude to a value correlated with optimal efficiency of forward movement.

**\*For correspondence:** samuel@ physics.harvard.edu (ADS); yzhang@oeb.harvard.edu (YZ)

[†]These authors contributed equally to this work

**Present address:** [‡]Graduate School of Science, Kobe University, Kobe, Japan

**Competing interests:** The authors declare that no competing interests exist.

## Introduction

In both vertebrates and invertebrates, neuromodulators profoundly shape the activity of neural circuits, giving rise to organized and yet flexible patterns of output (*Bargmann and Marder, 2013*). Neuromodulation of circuit activity is widespread in the neural circuits that generate structured locomotory patterns, such as walking, running, swimming, as well as feeding. Among a large number of neuromodulators, GABA, a major inhibitory neurotransmitter found in all nervous systems, plays a critical role in modulating locomotory patterns by regulating the underlying neural circuits at multiple levels (*Tegner et al., 1993*; *Cazalets et al., 1998*; *Reith and Sillar, 1999*; *Swensen et al., 2000*; *Ziskind-Conhaim, 2013*; *Fidelin et al., 2015*).

A great deal is known about the modulatory role of GABA in regulating locomotory patterns. In the crab *Cancer borealis*, GABA is identified in a few neuromodulatory neurons that project to the stomatogastric nervous system, which generates a pyloric rhythmic pattern. All the stomatogastric neurons respond to GABA with either an excitatory or inhibitory effect (*Swensen et al., 2000*), which are likely to contribute to the diverse motor patterns that can be generated by the stomatogastric system. The modulatory role of GABA has also been characterized in the spinal motor networks, where reciprocal glycinergic inhibition between contralateral and antagonist neuronal pools gives

rise to rhythmic motor activities in different vertebrate animals (*Alford and Williams, 1987*; *Brown, 1914*; *Buchanan, 1982*; *Cohen and Harris-Warrick, 1984*; *Dale, 1985*; *Friesen, 1994*; *Sharp et al., 1996*; *Soffe, 1987*). For example, in isolated lamprey spinal cord electrophysiological and pharmacological data showed that an internally released GABA signal regulated spinal burst rate and modulated intersegmental coordination (*Tegner et al., 1993*). In *Xenopus laevis* tadpoles, pharmacologically manipulating the activity of spinal GABA$_A$ receptor(s) alters the duration and rate of swimming episodes (*Reith and Sillar, 1999*). In zebrafish larva, the activity of a set of GABA-producing sensory neurons in the spinal cord inhibits the slow fictive swimming events during either resting or active state (*Fidelin et al., 2015*). In the in vitro spinal cord preparation isolated from neonatal rats, GABA modulates the locomotory patterns (*Cazalets et al., 1998*) and in the neonatal mouse spinal cord, GABAergic transmission is shown to play an integrated role in generating motor activity patterns (*Ziskind-Conhaim, 2013*).

While the role of GABA in shaping the activity patterns of motor networks is extensively studied, important questions remain to be addressed. For example, GABAergic neurons are often widely distributed in nervous systems to regulate diverse neuronal types. GABAergic neurotransmission can act either synaptically through the ionic GABA$_A$ receptors or extrasynaptically through metabotropic GABA$_B$ receptors (*Dittman and Regehr, 1997*; *Isaacson et al., 1993*; *Sigel and Steinmann, 2012*). Therefore, the high level of heterogeneity in the properties of GABA signals precludes a complete understanding of how GABA modulates the operation of a given neural circuit. In addition, while a large body of electrophysiological and pharmacological evidence revealed the compelling role of GABAergic modulation in motor activities, the causal link between the modulatory activity of GABA and the behavioral consequence in locomotion has not been established in behaving animals.

The nematode *Caenorhabditis elegans* provides an opportunity to address these questions. The connectivity of the 302 neurons in a hermaphrodite adult *C. elegans* is well-defined (*White et al., 1986*), allowing identification of individual neurons and their connectivity with the rest of the nervous system. The nervous system of the nematode is easily accessed by genetic tools, making it feasible to monitor and manipulate the activity of specific neurons in behaving animals in order to address the causal role of neural activity in movements. In addition, the small and fully sequenced genome encodes homologues of many known molecules that function in the mammalian brains, allowing characterization of these conserved factors in a genetically tractable organism (*Bargmann, 1998*).

Under the standard condition on a solid agar surface *C. elegans* generates forward undulatory movement that occurs at around 1 Hz bending oscillation on the dorsal-ventral plane. The undulatory locomotory wave travels posteriorly from head to tail through contraction of body muscles (*Wen et al., 2012*). In an adult hermaphrodite, two major motor networks mediate forward movement, the head motor circuit and the ventral nerve cord motor circuit that innervate 95 muscle cells arranged in parallel rows along its dorsal and ventral sides (*Altun and Hall, 2009*). The head motor circuit that innervates the anterior muscle cells in the head and neck consists of several groups of excitatory cholinergic motor neurons, including four SMD (sublateral motor neurons class D) neurons, and one group of inhibitory GABAergic motor neurons, the four RME (ring motor neuron class E) neurons (*Figure 1A*, *Altun and Hall, 2009*; *Gray et al., 2005*; *White et al., 1986*). The ventral nerve cord motor circuit that innervates the rest of the body wall muscles consists of excitatory cholinergic A-type and B-type motor neurons, which are required for backward and forward movement, respectively, as well as D-type GABAergic inhibitory motor neurons (*Chalfie et al., 1985*; *White et al., 1976*; *1986*). All ventral nerve cord motor neurons are subdivided into dorsal and ventral groups that innervate dorsal and ventral muscles, respectively; and both A- and B-type cholinergic neurons activate D-type GABAergic neurons that innervate the antagonistic muscles on the opposing side. This network connectivity allows alternating contraction and relaxation of the dorsal and ventral muscles that propagates the undulatory body wave from the head region to the rest of the body during forward movements (*Chalfie et al., 1985*; *Wen et al., 2012*; *White et al., 1976*, *1986*).

Previous studies showed that disrupting biosynthesis of GABA altered the amplitude of undulatory head bending without disrupting forward movement (*McIntire et al., 1993a*; *1993b*), suggesting that GABAergic neurotransmission is not required to generate forward undulatory body waves, but plays a modulatory role. In this study, we characterize the mechanism whereby GABA modulates the amplitude of head bending during forward movement. We show that the GABAergic head motor neurons RME and the cholinergic motor neurons SMD functionally interact

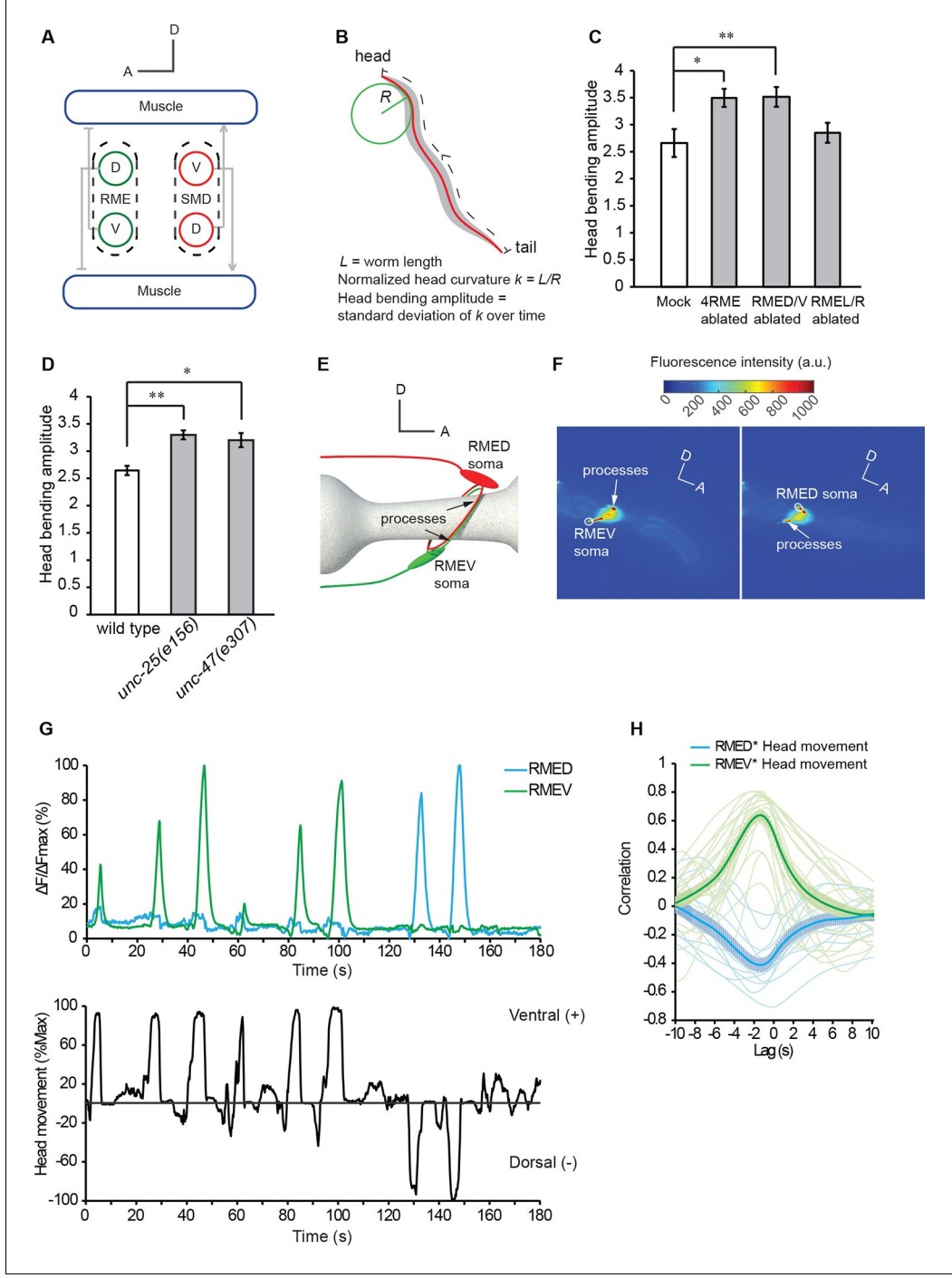

**Figure 1.** The GABAergic motor neurons RME restrict head bending amplitude and exhibit intracellular calcium signals that are correlated with head bending. (**A**) Schematics showing the innervation of anterior muscles by RME and SMD motor neurons. Note that the cell bodies (denoted by circles) of RMEV (V) and SMDD (D) are located on the ventral side and the cell bodies (denoted by circles) of RMED (D) and SMDV (V) are located on the dorsal side. RME and SMD neurons innervate the muscles that are contralateral to the position of their cell bodies. Arrows denote excitatory synapses and blunt-ended lines denote inhibitory synapses. Only the muscles and motor neurons on the left-side are shown (*White et al., 1986*). A, anterior; D, dorsal. (**B**) Schematics showing the method of quantifying the amplitude of head bending, which is defined as the standard deviation of head curvature along the first ~18% of the worm body over the time lapse of measurement (Materials and methods). (**C**) Ablating all 4 RME neurons or the dorsal and ventral RME (RMED/V) increases head bending amplitude, but ablating only the left and right RME neurons (RMEL/R) does not have an effect. (**D**) The *unc-25(e156)* and *unc-47(e307)* mutant

*Figure 1 continued on next page*

*Figure 1 continued*

animals exhibit increased head bending amplitude. For **C** and **D**, One-way ANOVA with Dunnett's post-test, \*\*$p<0.01$, \*$p<0.05$, $n \geq 9$ animals each, bar graphs indicate mean values and error bars indicate SEM. (**E**, **F**) Schematics showing the positions of RMED/V cell bodies and processes (**E**) and single frames of GCaMP3 fluorescence signals in RMED and RMEV (**F**). A, anterior; D, dorsal; a.u., arbitrary unit. (**G**) Sample GCaMP3 signals in RMEV and RMED neurons and the corresponding head bending in the same animal. *Figure 1—figure supplement 1.* shows samples of cross-correlation between the calcium transients in the cell body and neurite of a RMED neuron or a RMEV neuron. (**H**) Cross-correlation between RMEV or RMED GCaMP3 signal and head bending. Faint lines indicate the results from individual animals and the thick lines indicate mean value.
The following figure supplement is available for figure 1:

**Figure supplement 1.** Sample cross-correlation between the GCaMP3 signal in the cell body and the GCaMP3 signal in the neurite of a RMEV or a RMED neuron.

through extrasynaptic neurotransmission and conserved metabotropic receptors. The inhibitory feedback between RME and SMD provides a gain control mechanism to regulate the extent of head bending. The activity of RME plays a causal role in setting the amplitude of head bending to a value that correlates with optimal speed and propulsion efficiency of forward movement.

## Results

### RME motor neurons modulate head bending

During undulatory forward movement, *C. elegans* generates sinusoid body waves that oscillate at around 1 Hz on the dorsal-ventral plane. The undulatory body waves travel posteriorly from the head region (*Wen et al., 2012*). The anterior groups of muscle cells in the head and neck are innervated by several groups of cholinergic excitatory motor neurons, including four SMD neurons with two SMDD neurons innervating dorsal muscles and two SMDV neurons innervating ventral muscles (*Figure 1A*) (*Altun and Hall, 2009*; *Gray et al., 2005*; *McIntire et al., 1993b*; *White et al., 1986*). Similar groups of anterior muscle cells are also innervated by four GABAergic RME motor neurons, RMED (Dorsal), RMEV (Ventral), RMEL (Left), and RMER (Right) that innervate the contralateral muscles in the ventral, dorsal, right, and left quadrants, respectively (note that different from excitatory motor neurons, RMED innervates *ventral* muscles and RMEV innervate *dorsal* muscles) (*Figure 1A*) (*McIntire et al., 1993b*; *White et al., 1986*). To characterize the role of GABAergic neurotransmission in undulatory head bending, we focused on the RME GABAergic neurons. We first killed the four RME neurons with a laser beam with the aid of a fluorescent reporter transgene that was expressed in RME and other GABAergic neurons. We found that the operated animals still made undulatory head bending and forward movement, suggesting that the RME GABAergic neurons are not required for the generation of the movements. However, killing RME increased the amplitude of the undulatory head bending during forward movement (*Figure 1B and C*). This result, consistent with an earlier study (*McIntire et al., 1993b*), shows that the RME GABAergic neurons modulate the amplitude of the undulatory head bending.

To test specific roles among RME neurons, we ablated the RMED/V and RMEL/R subsets separately. We found that killing the RMED/V pair increased head bending amplitude along the dorsal-ventral axis to the same extent as ablating all four RME neurons, whereas killing RMEL/R did not generate any obvious defect (*Figure 1C* and *Videos 1*, *2*). Thus, RMED/V neurons negatively regulate head bending amplitude. Henceforth, RME refers to the RMED/V neurons in this study. Since the RME neurons are GABAergic, we tested the effect of removing components of GABA signaling pathway, including UNC-25, the GABA biosynthetic enzyme glutamic acid decarboxylase (GAD), and UNC-47, a vesicular transporter of GABA (*Eastman et al., 1999*; *Jin et al., 1999*; *McIntire et al., 1993a*). Both *unc-25(e156)* and *unc-47(e307)* mutants showed increased amplitude of head bending during forward movement (*Figure 1D* and *Videos 1*, *3*). Together, our results indicate that the RME GABAergic neurons modulate forward movement by regulating the extent of the undulatory head bending.

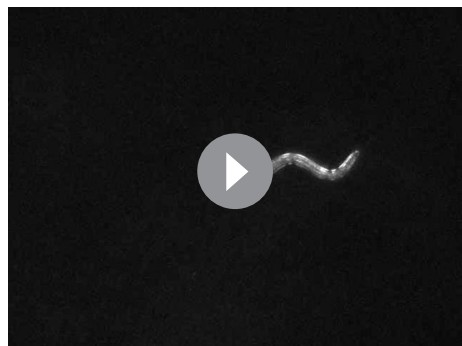

**Video 1.** Undulatory movement of a wild-type animal on an agar plate. The animal moves forward towards right at the beginning of the movie.

## RME neurons exhibit undulatory calcium activity

To understand how the RME neurons modulate head bending, we recorded the intracellular calcium activity of RME using transgenic animals expressing GCaMP3 (*Tian et al., 2009*) in all GABAergic neurons (*Punc-25::GCaMP3*). We used a microfluidic device in which an animal was able to bend its head in the dorsal-ventral direction (*Chronis et al., 2007*). Each RME neuron extends one process posteriorly and two contralaterally, and the contralaterally extending processes innervate head muscles (*Figure 1A,E and F*). We measured the calcium transients in the somata of RME neurons, since their somata and their muscle-innervating processes displayed correlated calcium signals under the experimental conditions (*Figure 1—figure supplement 1*). We found that RME generated undulatory calcium signals during head bending: RMED displayed increased intracellular calcium transients during dorsal head bending and RMEV displayed increased intracellular calcium transients during ventral head bending (*Figure 1F,G* and *Video 4*). We defined ventral head bending as positive and dorsal head bending as negative. Using cross-correlation analysis, we found that RMEV calcium activity positively correlated with head bending and RMED calcium activity negatively correlated with head bending (*Figure 1H*). The calcium signals in RMEV and RMED were anti-correlated during head bending. The oscillatory activity pattern allows RME to regulate head bending with a temporal pattern that matches that of head undulation.

## The head bending-correlated activity of RME is driven by cholinergic neurotransmission

Next, we asked how the undulatory activity of RME was regulated. We first examined whether the head bending-correlated RME calcium activity might be attributable to a proprioceptive response to head bending. We found that in animals immobilized with microbeads (*Kim et al., 2013*) RMED and REMV neurons generated oscillatory and anti-correlated calcium signals (*Figure 2—figure supplement 1*). In addition, we observed similar calcium dynamics in the RME neurons in the *unc-54(e1092)* null mutants (*Figure 2—figure supplement 2*), which lacked a major myosin heavy chain protein and were defective in muscle contraction (*Dibb et al., 1985*; *MacLeod et al., 1977*). Therefore, the undulatory calcium activity of the RME GABAergic neurons does not require head movement.

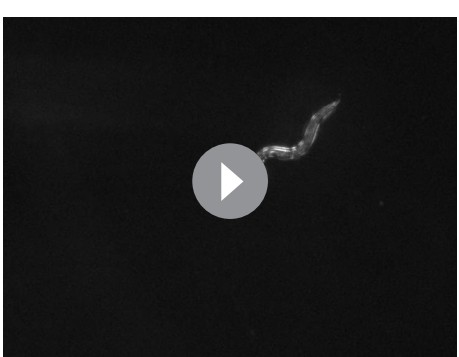

**Video 2.** Undulatory movement of a wild-type animal with RMED and RMEV ablated. The animal in the movie moves towards lower-left corner on an agar plate at the beginning of the movie. Note the increased head bending amplitude in the animal.

To characterize the regulation of RME, we sought the type of neurotransmission that was required for the head-bending correlated activity in RME. We first examined *unc-13(e51)* mutants, which were defective in neurotransmitter release (*Brenner, 1974*; *Lackner et al., 1999*; *Miller et al., 1996*; *Richmond et al., 1999*). We found that the *unc-13* mutant animals bent their heads in the microfluidic imaging device and generated active calcium transients in RME. However, the RME calcium activity in the *unc-13* mutants did not correlate with head bending (*Figure 2A and B*). In contrast, blocking neuropeptide release in the *unc-31(e928)* (*Ann et al., 1997*; *Avery et al., 1993*) mutants did not have an obvious effect on RME calcium activity (*Figure 2A and B*). These results indicate that neurotransmitters, but not neuropeptides,

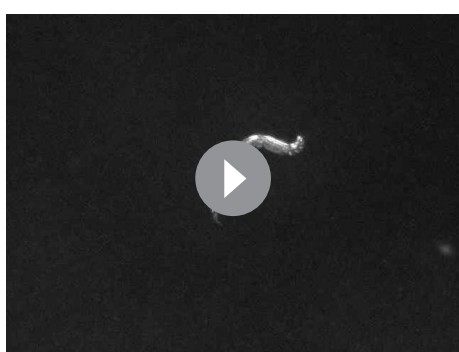

**Video 3.** Undulatory movement of an *unc-25(e156)* mutant animal on an agar plate. The animal moves towards upper-right corner at the beginning of the movie and then turns to move towards lower-right corner. Note the increased head bending amplitude in the *unc-25* mutant animal.

regulate head bending-correlated calcium dynamics in RME.

Next, we sought the type of neurotransmitter that underlies RME activity. Because RME are post-synaptic to several neurons that are cholinergic (*Duerr et al., 2008*; *White et al., 1986*), we performed calcium imaging in mutants that lacked the choline acetyltransferase CHA-1, which is required for biosynthesis of acetylcholine and expressed in all cholinergic neurons (*Alfonso et al., 1994*; *Rand and Russell, 1984*). We found that similar to the *unc-13(e51)* mutants the hypomorphic allele *cha-1(p1152)* (*Rand and Russell, 1984*) also exhibited active calcium transients in RME that were not correlated with head bending (*Figure 2A and B*). The defect in RME calcium activity in the *cha-1 (p1152)* mutants was rescued by expressing the cosmid ZC416 that contained the genomic locus of *cha-1* (*Figure 2C and D*). These results demonstrate that the head bending-correlated calcium activity in RME depends on cholinergic neurotransmission.

Next, we sought the cholinergic neurons that regulated RME calcium activity. Because SMB and IL2 are major presynaptic cholinergic neurons of RME, we selectively expressed the full-length *cha-1* cDNA in either SMB or IL2 neurons in the *cha-1(p1152)* mutant animals and examined the resulting effects on RME calcium activity. Surprisingly, we found that restoring *cha-1* expression in the SMB motor neurons with the *odr-2(18)* promoter (*Chou et al., 2001*) or in the IL2 neurons with the *klp-6* promoter (*Lee et al., 2011*; *Peden and Barr, 2005*) could not rescue the defects in the head-bending correlated calcium activity in the RME neurons in the *cha-1(p1152)* mutant animals (*Figure 2E and F*). These results suggest that the head bending-correlated calcium activity in RME may be independent of synaptic inputs.

## The SMD motor neurons regulate RME through extrasynaptic neurotransmission

To identify the cholinergic neurons that drive the undulatory calcium activity in RME, we examined the motor neurons SMD, because we previously showed that similar to RME the cholinergic SMDD and SMDV motor neurons that innervated the same sets of head and neck muscles also displayed calcium activity that was correlated with dorsal-ventral head bending (*Figure 1A*) (*Hendricks et al., 2012*). We first generated transgenic animals that allowed us to simultaneously monitor calcium activities in SMD and RME with the calcium sensitive fluorescent reporter GCaMP3. We found that SMDV and RMEV generated increased calcium transients during ventral head bending, while SMDD and RMED generated increased calcium transients during dorsal head bending (*Figure 3A–C* and *Video 5*). The calcium activity of SMD correlated with that of RME during head bending and led

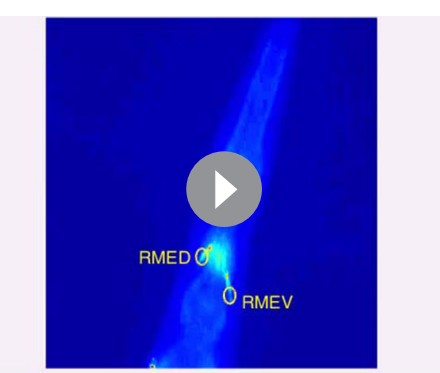

**Video 4.** A representative movie for GCaMP3 signals in RMED and RMEV neurons and the head bending in a transgenic animal that expresses GCaMP3 in GABAergic neurons. Both REMD and RMEV extend processes to the contralateral side to innervate the contralateral muscles and these processes run side-by-side. Because the calcium signals in the cell body and process of a RME neuron correlate with each other, we measure the calcium dynamics in the cell bodies of RME in this study. The RMED or RMEV cell body is highlighted with a circle in the movie. Please refer to *Figure 1E and F* for the positions of RME cell bodies and processes. Ventral to the right and anterior to the upper-right corner.

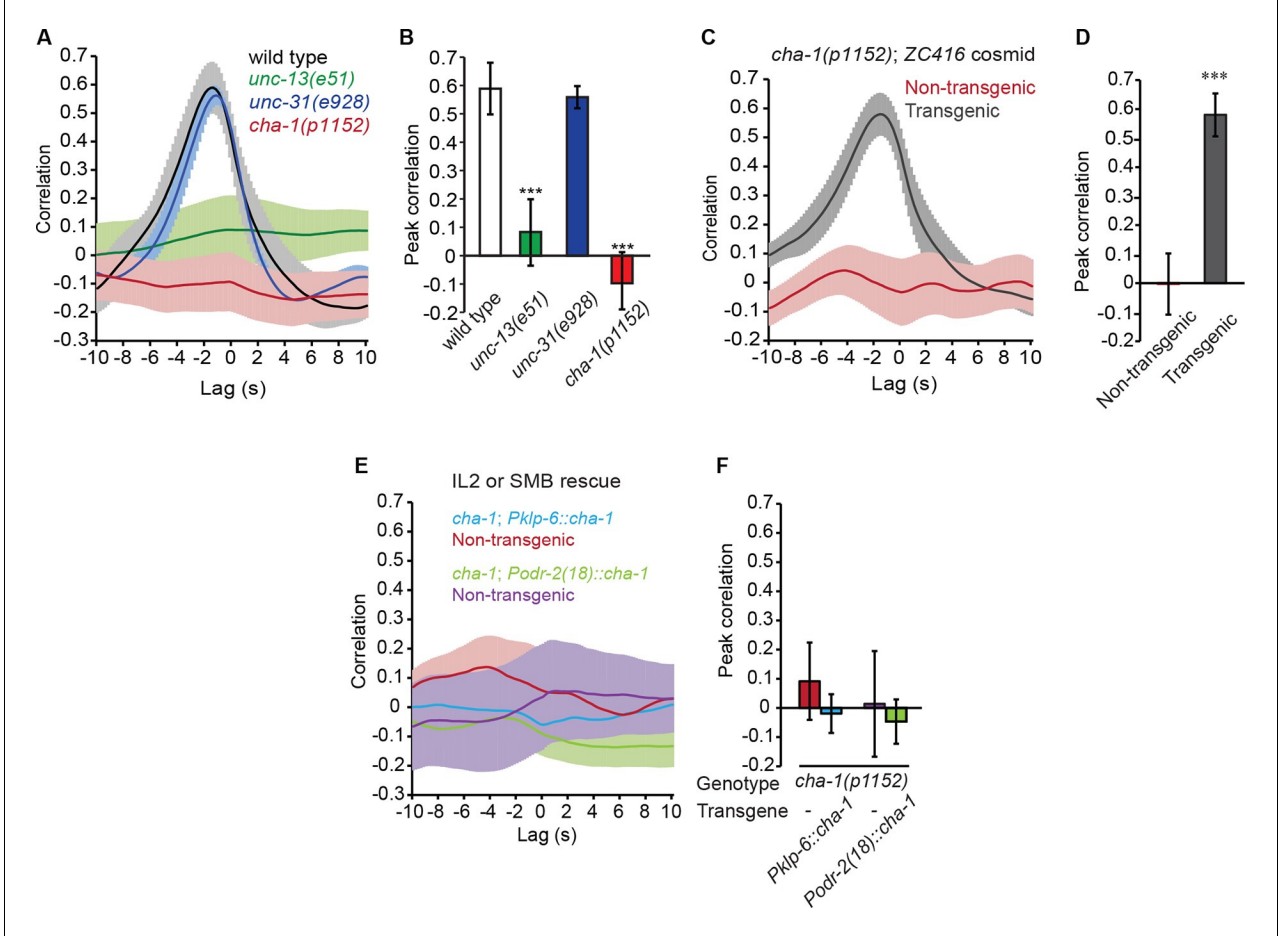

**Figure 2.** The head bending-correlated calcium activity of RME depends on cholinergic neurotransmission. (A, B) The *unc-13(e51)* and *cha-1(p1152)* mutant animals are significantly defective in cross-correlation (A) and peak correlation (B) between RME calcium activity and head bending; but the *unc-31(e928)* mutant animals are normal. *Figure 2—figure supplement 1* and *2* show the representative GCaMP3 signals in RME neurons in immobilized and in *unc-54(e1092)* animals, respectively. (C, D) The defects of *cha-1(p1152)* mutant animals in cross-correlation (C) and peak correlation (D) between RME calcium activity and head bending is rescued by cosmid ZC416 that contains the coding region of *cha-1* genomic DNA. (E, F) Expressing a wild-type *cha-1* cDNA in IL2 (*Pklp-6::cha-1*) or SMB (*Podr-2(18)::cha-1*) does not rescue the correlation (E) or peak correlation (F) between RME calcium activity and head bending in *cha-1(p1152)* mutant animals. For B, one-way ANOVA with Dunnett's post-test. For D and F, transgenic animals are compared with nontransgenic siblings with student's *t*-test. For all, ***p<0.001, n ≥ 7 animals each, Mean ± SEM, peak correlation is the highest correlation within the 1 s time window centered on the peak correlation of the wild-type control in A and B; while similar effects were usually observed in more than one transgenic lines, the effect of each transgene is reported with the results from one transgenic line.

The following figure supplements are available for figure 2:

**Figure supplement 1.** Representative traces for GCaMP3 signal in RME in animals immobilized with microbeads (Materials and methods).

**Figure supplement 2.** Representative traces for calcium signal in RME in *unc-54(e1092)* mutant animals.

that of RME (*Figure 3A–C*). These results support the possibility that the cholinergic transmission of SMD regulates RME activity. Next, to address this hypothesis, we expressed the *cha-1* cDNA under the *glr-1* or the *lad-2* promoter in the *cha-1(p1152)* mutant animals and found that either *Pglr-1::cha-1* or *Plad-2::cha-1* fully rescued the RME calcium dynamics that was correlated with dorsal-ventral head bending (*Figure 3D and E*). Both the *glr-1* and *lad-2* promoters drive expression in multiple cells, but the only cholinergic head motor neurons that consistently express both of the promoters are SMD (*Brockie et al., 2001*; *Hendricks et al., 2012*; *Wang et al., 2008*). To strengthen the specificity of our results, we ablated SMD neurons in the transgenic animals that expressed *Pglr-1::cha-1* in the *cha-1(p1152)* mutant animals. We found that the rescuing effect on the RME calcium activity

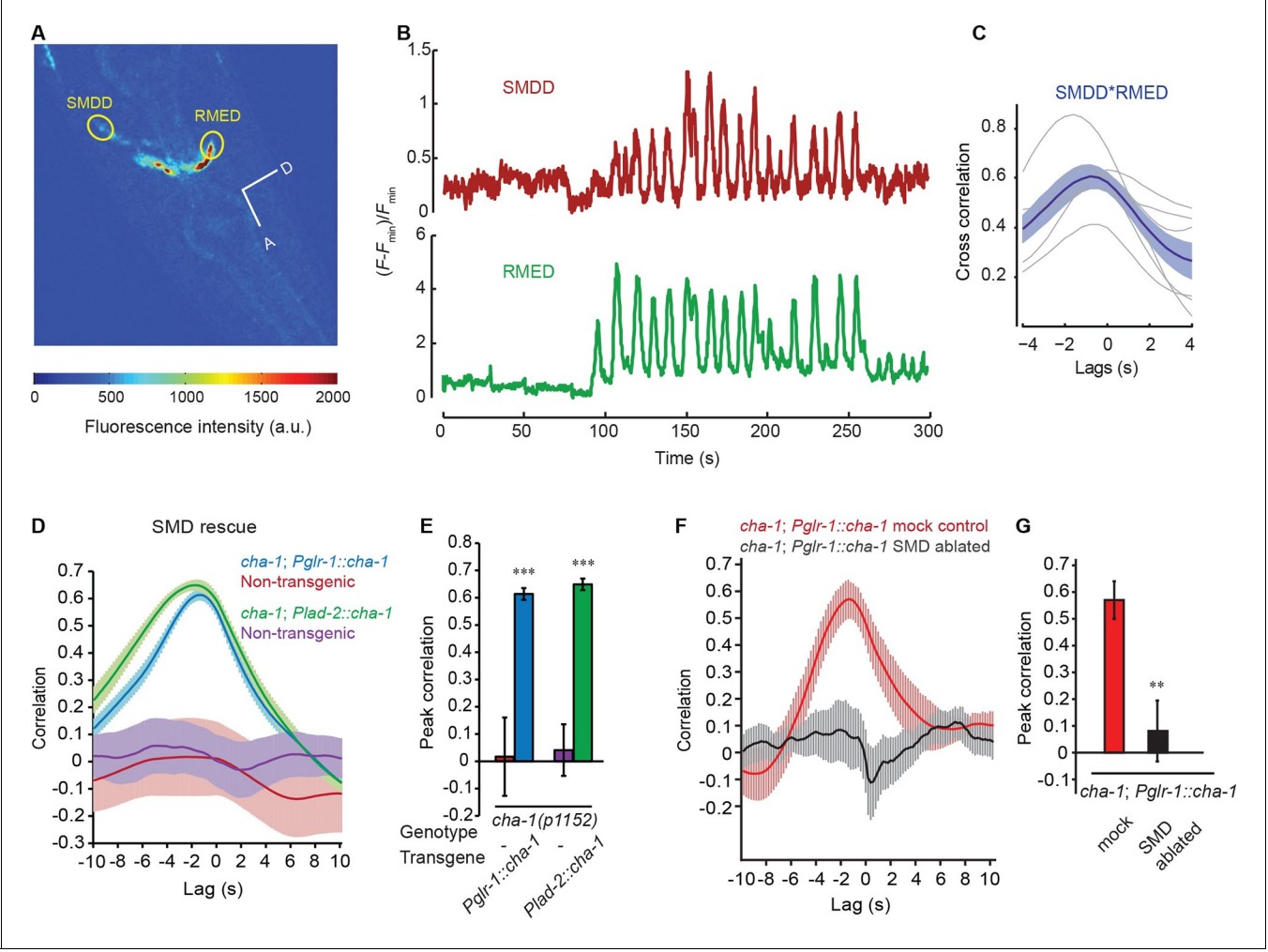

**Figure 3.** The head bending-correlated calcium activity of RME is regulated by the cholinergic signal from SMD. (**A**) Volumetric view of a 3-D image stack from an animal expressing *Pglr-1::GCaMP3* and *Punc-25::GCaMP3*. A, anterior; D, dorsal; circles highlight cell bodies; a.u., arbitrary unit. (**B**) Sample traces of calcium dynamics in SMDD and RMED during head movement. (**C**) Cross-correlation between calcium activities in SMDD and RMED. $n = 6$ animals, faint lines indicate the results from individual animals, the thick line indicates the mean value and the shade indicates SEM. (**D**, **E**) Expressing a wild-type *cha-1* cDNA in SMD neurons (*Pglr-1::cha-1* or *Plad-2::cha-1*) rescues the head-bending correlated calcium activity in RME. Peak correlation is the highest correlation within the 1 s time window centered on the peak correlation of the wild-type control in *Figure 2A and B*. (**F**, **G**) Ablating SMD in the transgenic animals that express the wild-type *cha-1* cDNA with the *glr-1* promoter in the *cha-1(p1152)* mutant animals abolishes the rescuing effect on the head-bending correlated calcium activity in RME. Peak correlation is the highest correlation within the 1 s time window centered on the peak correlation of the mock control. For (**E** and **G**), transgenic animals are compared with non-transgenic siblings with student's *t*-test. For (**D-G**) ***p<0.001, **p<0.01, $n \geq 5$ animals each, Mean ± SEM; while similar effects were usually observed in more than one transgenic lines, the effect of each transgene is reported with the results from one transgenic line.

was completely lost in the ablated animals (*Figure 3F and G*). Together, these results support the instructive role of SMD in generating the oscillatory calcium activity of RME.

Intriguingly, SMD neurons do not synapse onto RME (*White et al., 1986*), suggesting that extrasynaptic neurotransmission from SMD regulates RME. To characterize the potential involvement of extrasynaptic cholinergic neurotransmission, we sought the cholinergic receptor that mediated the neurotransmission from SMD to RME. Extrasynaptic neurotransmission is regulated by G-protein coupled receptors, because the high binding affinity of these receptors allows intercellular signaling over a relatively long distance (*Hille, 1992*). *C. elegans* has three G-protein coupled acetylcholine receptors that are encoded by *gar-1, gar-2* and *gar-3* and are similar to the mammalian muscarinic acetylcholine receptors (*Hwang et al., 1999*; *Lee et al., 2000*; *Park et al., 2000*). Previously, it was

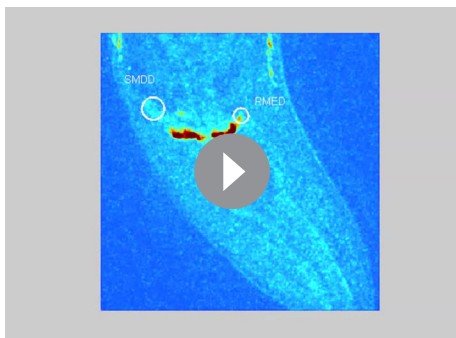

**Video 5.** A representative movie for GCaMP3 signals in RMED/V and SMDD/V neurons and the head bending in a transgenic animal that expresses GCaMP3 in a few neurons, including RMED/V and SMDD/V. RMED and SMDD neurons are labeled with circles on the movie frames. Dorsal to the right and anterior to the low-right corner.

shown that *gar-2* was widely expressed in the nervous system, including in the GABAergic motor neurons (*Dittman and Kaplan, 2008*; *Lee et al., 2000*). Interestingly, we found that the head bending-correlated activity of RME was significantly reduced in the deletion mutant *gar-2(ok520)* and the defect was fully rescued by expressing a wild-type *gar-2* DNA fragment (*Figure 4A and B*). This result indicates that GAR-2 mediates the SMD cholinergic signal to generate the undulatory activity pattern of RME. The reduced activity of the RME neurons in the *gar-2(ok520)* mutants also predicts that the *gar-2(ok520)* animals should exhibit increased head bending amplitude. Indeed, we found that *gar-2 (ok520)* displayed significantly larger amplitude of head bending (*Figure 4C* and *Video 6*), similar to the effect of removing RMED and RMEV with laser ablation (*Figure 1C*). Expressing a wild-type *gar-2* cDNA (*Dittman and Kaplan, 2008*) with either the endogenous *gar-2* promoter or the *unc-25* promoter that was selective for the GABAergic neurons (*Jin et al., 1999*) fully rescued the defect of the *gar-2(ok520)* mutants in head bending amplitude (*Figure 4C*). Among all GABAergic neurons only RME innervate head muscles and regulate head bending (*Figure 1C*) (*McIntire et al., 1993b*; *White et al., 1976*; *1986*). Taken together, our results demonstrate that the extrasynaptic cholinergic neurotransmission of SMD facilitates the oscillatory activity of RME via the muscarinic acetylcholine receptor GAR-2 to regulate the amplitude of head bending. To exclude the possibility that indirect synaptic neurotransmission underlies the modulatory effect of the SMD cholinergic signal on RME, we further examined the synaptic wiring (*White et al., 1986*). We showed that ablating SMD completely removed the rescuing effect of *Pglr-1::cha-1* on the undulatory activity of RME in the *cha-1(p1152)* mutant animals (*Figure 3F and G*). We found that among all the neurons that consistently express the *glr-1* promoter, only the cholinergic head neurons RMDD and RMDV are postsynaptic of SMD or connect with SMD via gap junction. However, neither RMDD nor RMDV neurons synapse onto RME. Overall, our results in combination with the connectome reveal that the head-bending correlated activity of RME is generated by extrasynaptic cholinergic neurotransmission that originates from the head motor neurons SMD.

To further characterize the function of the SMD cholinergic neurons in driving RME calcium activity, we blocked the neurotransmission from SMD in wild-type animals by expressing Tetanus Toxin (TeTx) (*Schiavo et al., 1992*) with either the *glr-1* or *lad-2* promoter. We found that either mutation disrupted the head bending-correlated calcium dynamics in RME (*Figure 4D and E*). However, blocking neurotransmission by expressing TeTx in two sets of major presynaptic neurons of RME, the cholinergic motor neurons SMB (*Podr-2(18)::TeTx*) or sensory neurons IL2 (*Pklp-6::TeTx*), or both SMB and IL2 had no effect on RME calcium activity (*Figure 4—figure supplement 1*), indicating that general reduction in cholinergic neurotransmission is not sufficient to disrupt RME calcium dynamics. To address the question more specifically, we also performed laser ablation of SMD. We found that ablating SMD disrupted head bending-correlated calcium activities in RME while the RME neurons in the SMD-ablated animals remained active (*Figure 4F and G*). In contrast, ablating the IL2 sensory neurons had no effect (*Figure 4F and G*). Taken together, these results demonstrate that the cholinergic excitatory motor neurons SMD drive the head bending-correlated oscillatory activity in RME through an extrasynaptic signal. RME generates a modulatory signal that limits head bending amplitude. The oscillatory activity of RME provides a mechanism through which the GABAergic modulatory signal mediates head bending with a temporal pattern that matches that of head undulation.

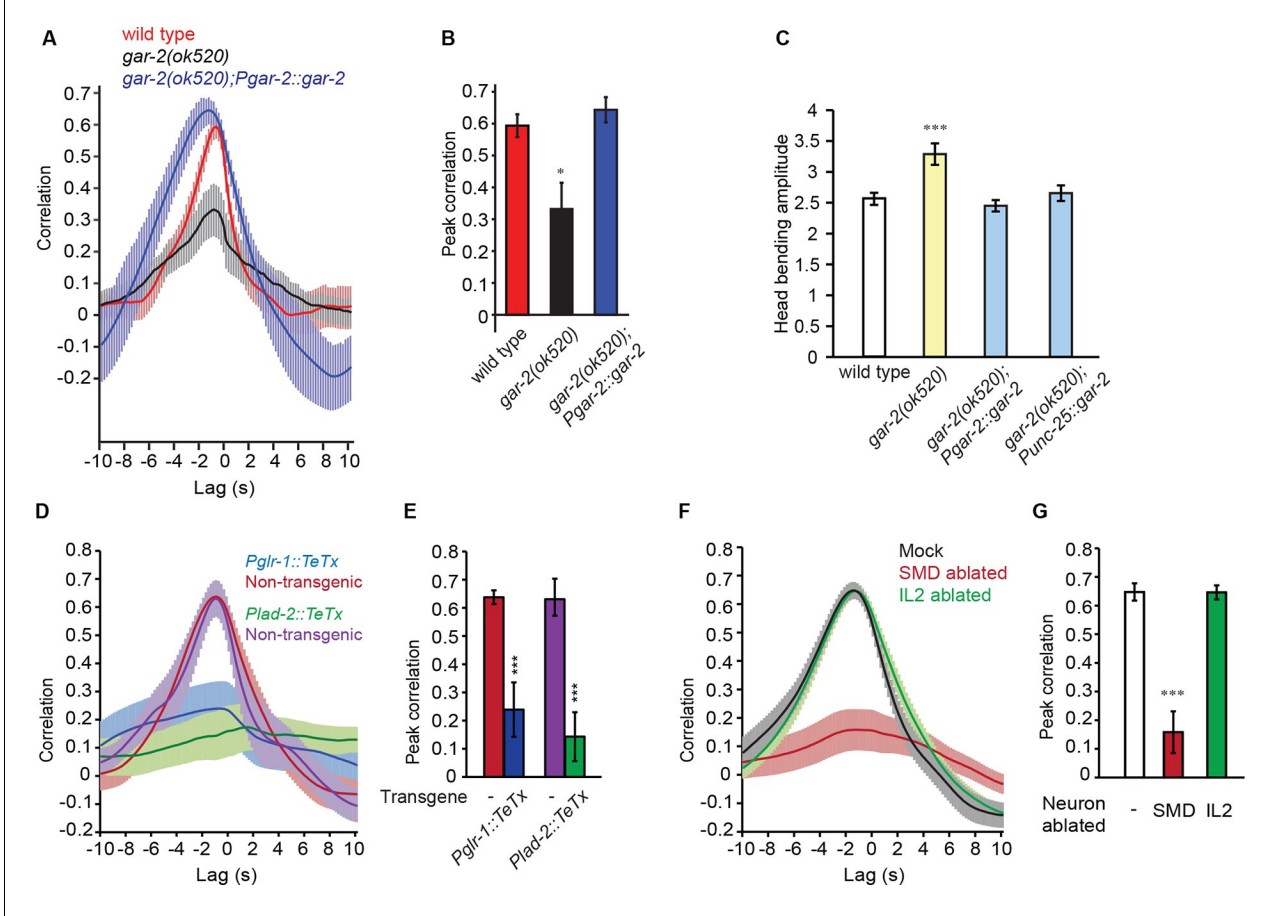

**Figure 4.** SMD regulate the activity of RME via the metabotropic acetylcholine receptor GAR-2. (**A**, **B**) The *gar-2(ok520)* mutants are defective in cross-correlation (**A**) and peak correlation (**B**) between RME calcium activity and head bending, and the expression of *Pgar-2::gar-2* rescues the defects. Transgenic animals and mutants are compared with wild type with ANONA with Dunnett's post-test. Peak correlation is the highest correlation within the 1 s time window centered on the peak correlation of the wild-type control. $n \geq 7$ animals each. (**C**) A deletion mutation in *gar-2(ok520)* significantly increases the amplitude of head bending and the defect is rescued by expressing a wild-type *gar-2* cDNA with the *gar-2* promoter (*Pgar-2::gar-2*) or with the *unc-25* promoter that is selectively expressed in GABAergic neurons (*Punc-25::gar-2*). *gar-2(ok520)* mutants and transgenic animals are compared with wild type with ANONA with Dunnett's post-test, $n \geq 9$ animals each. (**D**, **E**) Blocking neurotransmission from SMD neurons (*Pglr-1::TeTx* or *Plad-2::TeTx*) generates significant defects in cross-correlation (**D**) and peak correlation (**E**) between RME calcium activity and head bending. Transgenic animals are compared with non-transgenic siblings with student's t-test, $n \geq 9$ animals each. *Figure 4—figure supplement 1* shows that blocking neurotransmission from IL2 (*Pklp-6::TeTx*) or SMB (*Podr-2(18)::TeTx*) neurons or both does not alter the cross-correlation or peak correlation between RME calcium activity and head movement. (**F**, **G**) Ablating SMD, but not IL2, generates significant defects in cross-correlation (**F**) and peak correlation (**G**) between RME calcium activity and head bending. Ablated animals and mock controls are compared with student's t-test, $n \geq 9$ animals each. For **E** and **G**, peak correlation is the highest correlation within the 1 s time window centered on the peak correlation of the wild-type control in *Figure 2A and B*. For all, ***$p<0.001$, *$p<0.05$, Mean ± SEM; while similar effects were usually observed in more than one transgenic lines, the effect of each transgene is reported with the results from one transgenic line.

The following figure supplement is available for figure 4:

**Figure supplement 1.** Blocking neurotransmission from IL2 neurons (*Pklp-6::TeTx*) or SMB neurons (*Podr-2(18)::TeTx*) or both does not significantly alter the cross-correlation or peak correlation between RME calcium activity and head movement.

## The GABA$_B$ receptor GBB-1/2 functions in SMD neurons to restrain head bending

Next, to understand how RME regulate the amplitude of head undulation, we sought the downstream GABAergic receptors. We focused on the metabotropic GABA receptors, due to the modulatory role of RME on head bending (*Figure 1*). The *C. elegans* genome encodes the homologs of the subunits of

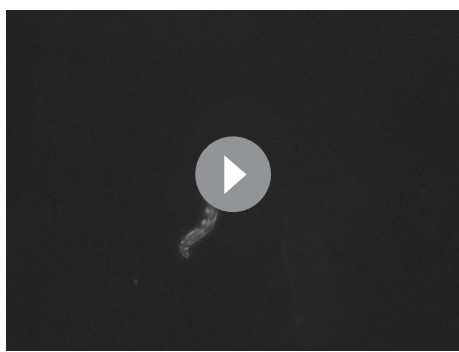

**Video 6.** Undulatory movement of a *gar-2(ok520)* mutant animal on an agar plate. The animal moves down at the beginning of the movie and then turns to move towards left and then up. Note the increased head bending amplitude in the *gar-2* mutant animal.

the mammalian metabotropic B-type GABAergic receptors GABA_BR1 and GABA_BR2, which are GBB-1 and GBB-2, respectively. Similar to their mammalian homologs GBB-1 and GBB-2 act as obligate heterodimers (*Dittman and Kaplan, 2008*). We first tested a deletion mutant *gbb-1 (tm1406)*. We found that similar to removing the GABA biosynthetic enzyme glutamic acid decarboxylase in the *unc-25(e156)* null mutant animals, the deletion mutation in *gbb-1(tm1406)* increased the amplitude of head bending during forward movement (*Figure 5A*). In addition, the double mutant *gbb-1(tm1406);gbb-2(tm1165)* similarly increased head bending amplitude (*Figure 5—figure supplement 1*), consistent with the notion that GBB-1 and GBB-2 function as heterodimers (*Dittman and Kaplan, 2008*). The exaggerated head bending in the *gbb-1(tm1406)* mutant animals was rescued by expressing a wild-type *gbb-1* cDNA under the *gbb-1* endogenous promoter (*Figure 5B*). Thus, similar to the RMED/V neurons (*Figure 1C*), the GABA_B receptor GBB-1/GBB-2 restrains the head bending amplitude.

Next, we sought the functional site of GBB-1. Previously, it was shown that a GFP reporter driven by the *gbb-1* promoter was expressed in the cholinergic neurons along the ventral nerve cord, but not in the GABAergic neurons or muscles (*Dittman and Kaplan, 2008*). We found that the transcriptional reporter of *gbb-1* was also expressed in a number of head neurons, including SMD (*Figure 5C–H*). Importantly, expressing the wild-type *gbb-1* cDNA with the *glr-1* or *lad-2* promoter restored normal head bending in the *gbb-1(tm1406)* mutant animals (*Figure 5I and J*). Among the head motor neurons, the expression of *Pglr-1::gbb-1* and *Plad-2::gbb-1* consistently overlaps in the SMD neurons (*Brockie et al., 2001*; *Hendricks et al., 2012*; *Wang et al., 2008*). Together, these results indicate that the GABA_B receptor acts in SMD to limit the amplitude of head bending and suggest that RME restrict head bending by negatively regulating SMD. This conclusion predicts that inhibiting SMD should generate smaller head bending. Consistently, we found that ablating SMD neurons significantly reduced the amplitude of head bending, indicating that SMD promote the amplitude of head bending (*Figure 5K* and *Video 7*). Together, these results indicate that the RME GABAergic neurotransmission limits the amplitude of head bending by inhibiting SMD, which positively regulate head undulation. Interestingly, there is only one synapse identified from RMED and RMEV to the four SMD neurons [RMEV→SMDD<u>R</u>(Right)] (*White et al., 1986*). Therefore, it is conceivable that the RME GABAergic neurons regulate SMD through extrasynaptic neurotransmission via the G-protein coupled GABAergic receptor GBB-1/GBB-2.

## Optogenetic analysis of the RME GABAergic modulatory signal

Our results propose that during head undulation a cholinergic signal from SMD neurons, which exhibit head bending-correlated activity and facilitate head bending, drives the oscillatory activity of the GABAergic RME motor neurons that subsequently inhibit SMD to restrain head bending. RME and SMD signal to each other through extrasynaptic neurotransmission via the muscarinic acetylcholine receptor GAR-2 and the metabotropic GABAergic receptor GBB-1/GBB-2. This small circuit provides a real-time modulatory mechanism whereby the head bending can be regulated during the movement to generate a precisely controlled locomotory pattern.

To further test this model and the causal role of the modulatory function of the RME GABAergic signal in regulating head bending, we turned to optogenetics. We acutely manipulated the activity of RME in moving animals by inhibiting or activating it with optogenetics and measured the resulting behavioral effects. We first tested the effect of inhibiting RME using transgenic animals that expressed the light-sensitive proton pump archaerhodopsin (Arch) in all GABAergic neurons (*Chow et al., 2010*; *Okazaki et al., 2012*). Using an illumination system, CoLBeRT, that tracked the position of a moving animal and illuminated selected regions on the worm body [Materials and methods and (*Leifer et al., 2011*)], we followed movement of freely-moving animals and selectively

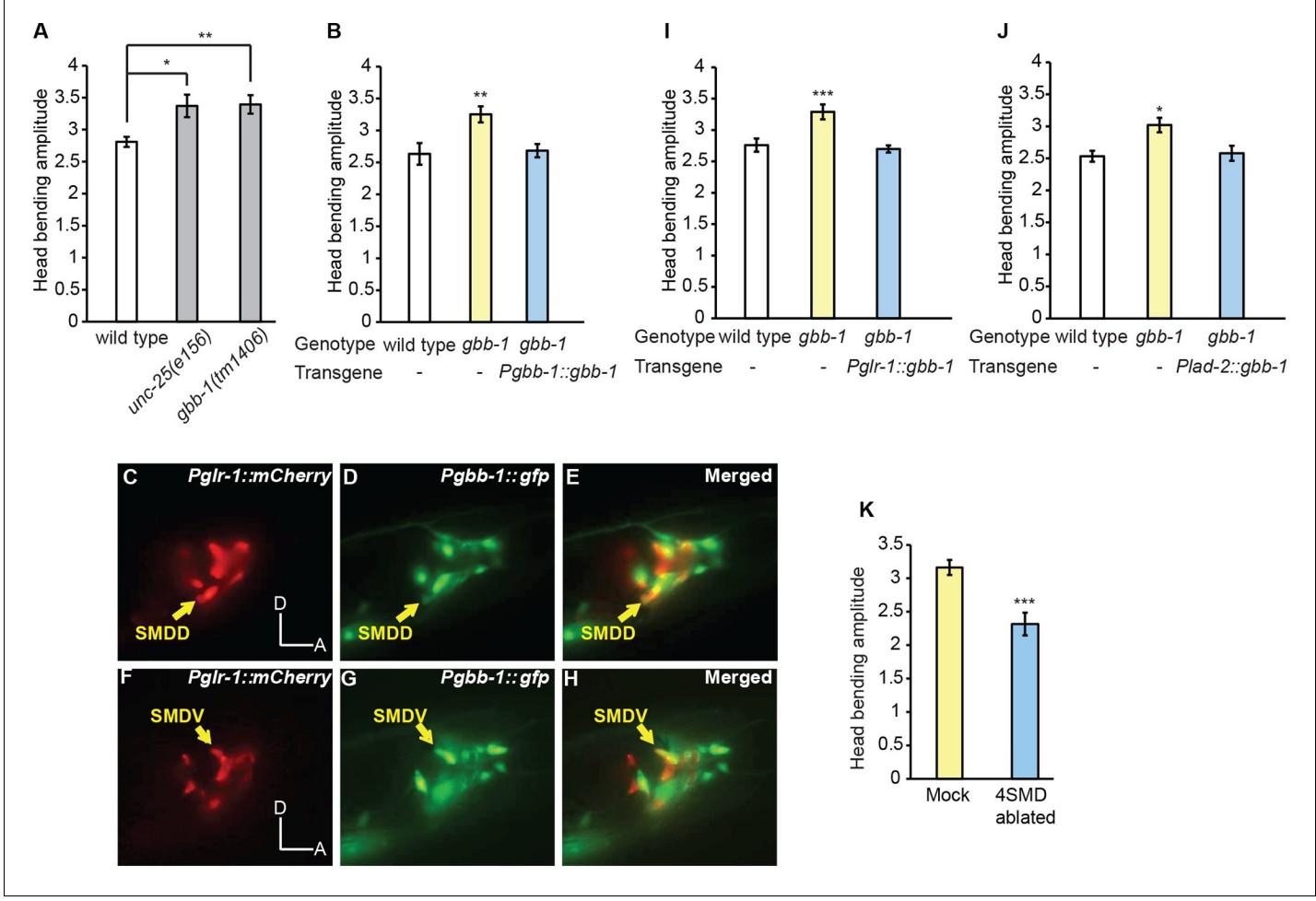

**Figure 5.** The GABA$_B$ receptor subunit GBB-1 acts in the SMD neurons to limit head bending amplitude. (**A**) The *gbb-1(tm1406)* mutants show an increased head bending amplitude similar to *unc-25(e156)* mutants. One-way ANOVA with Dunnett's post-test, *n* ≥ 8 animals each. *Figure 5—figure supplement 1.* shows the increased head bending amplitude in the *gbb-1;gbb-2* double mutant animals in comparison with wild type. (**B**) The exaggerated head bending in *gbb-1(tm1406)* mutants is rescued by expressing a wild-type *gbb-1* cDNA under the endogenous promoter of *gbb-1*. Transgenic animals (n=16 animals) are compared with non-transgenic siblings (n=15 animals) with student's *t*-test. (**C-H**) *gbb-1* is expressed in head neurons, including SMD. *Pglr-1::mCherry* is expressed in SMD and several other neurons. The expression of *Pgbb-1::gfp* and *Pglr-1::mCherry* overlap in SMD. Arrows denote SMDD or SMDV neuron. A, anterior; D, dorsal. (**I, J**) The exaggerated head bending in *gbb-1(tm1406)* mutants is rescued by expressing a wild-type *gbb-1* cDNA under the *glr-1* promoter (**I**) or the *lad-2* promoter (**J**). Transgenic animals (*n* ≥ 14 animals) are compared with non-transgenic siblings (*n* ≥ 13 animals) with student's *t*-test. (**K**), SMD-ablated animals show decreased head bending amplitude. Neuron-ablated animals are compared with mock controls with Student's *t*-test, n = 9 animals each. For all, ***p<0.001, **p<0.01, *p<0.05, Mean ± SEM; while similar effects were usually observed in more than one transgenic lines, the effect of each transgene is reported with the results from one transgenic line.

The following figure supplement is available for figure 5:

**Figure supplement 1.** The *gbb-1(tm1406);gbb-2(tm1165)* double mutant animals show increased head bending amplitude, similarly as the *gbb-1 (tm1406)* single mutants.

illuminated RME neurons with a green laser. We found that during forward movement optogenetic inhibition of RME caused exaggerated head deflection, mimicking the effect of RME ablation (*Figure 6A,C* and *Video 8*). The head bending amplitude returned to normal upon the removal of green light (*Video 8*). Next, we examined the effect of RME activation in transgenic animals expressing the light-gated ion channel channelrhodopsin (ChR2) (*Boyden et al., 2005*; *Liu et al., 2009*; *Nagel et al., 2003*) in all the GABAergic neurons. In these experiments, we ablated a GABAergic interneuron RIS, because the process of RIS was adjacent to the soma of RME and optogenetic activation of RIS was shown to induce sleep-like quiescence (*Turek et al., 2013*), potentially interfering

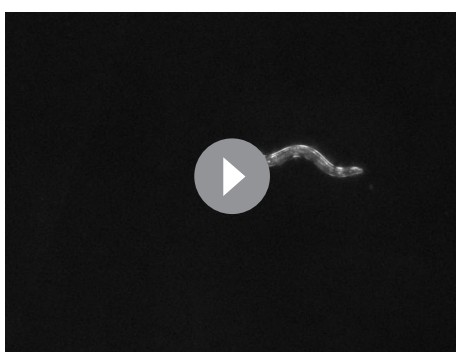

**Video 7.** Undulatory movement of a wild-type animal with SMDD and SMDV neurons ablated. The animal moves towards lower-right corner on an agar plate at the beginning of the movie. Note the reduced head bending amplitude in the animal.

with the behavioral effects of activating RME. Ablating RIS by itself does not have a significant impact on the amplitude of head bending (*McIntire et al., 1993b*). We found that optogenetic activation of RME, which was achieved by selective illumination of blue light with the CoLBeRT system [Materials and methods (*Leifer et al., 2011*)], caused a significant reduction in the amplitude of head bending, mimicking the effect of SMD ablation (*Figure 6B,C* and *Video 9*). Head bending amplitude returned to normal after removal of the blue light (*Video 9*). These results together indicate that inhibiting or activating RME in moving animals causally lead to increased or decreased head bending amplitude, respectively.

Next, we further characterized the behavioral effect of RME activation using mutants that were defective in GABA signaling. First, we found that optogenetic stimulation of RME in *unc-25(e156)* mutants did not change head bending amplitude, indicating that the behavioral effect of activating RME depends on GABA (*Figure 6C*). Second, activating RME in *unc-49(e407)*, a loss-of-function mutant of GABA$_A$ receptor that is expressed primarily in muscles (*Bamber et al., 1999*), still reduced head bending amplitude similarly as in wild type (*Figure 6C*), suggesting that the UNC-49 GABA$_A$ receptor is not critical for RME to limit the amplitude of head bending. In comparison, activating RME in the *gbb-1(tm1406)* mutant animals significantly reduced the effect of activating RME on head bending amplitude (*Figure 6C*) and activating RME in the *unc-49(e407);gbb-1(tm1406)* double mutant animals completely abolished the effect of RME activation on head bending amplitude (*Figure 6C*). Furthermore, expressing a wild-type *gbb-1* cDNA with either the *glr-1* or *lad-2* promoter rescued the effects of activating RME on head bending amplitude in the *unc-49(e407);gbb-1(tm1406)* mutant animals (*Figure 6C*). Expression of the *glr-1* and *lad-2* promoters consistently overlaps in the SMD neurons (*Brockie et al., 2001*; *Hendricks et al., 2012*; *Wang et al., 2008*). These results show that the GABA$_B$ receptor GBB-1/GBB-2 plays a critical role in mediating the modulatory effect of RME neurons on head bending amplitude and reveal a potential implication of the GABA$_A$ receptor in head bending regulation through an unknown mechanism. Taken together, these results demonstrate that RME neurons negatively regulate the amplitude of head bending through the GABA$_B$ receptor GBB-1/GBB-2 in SMD (*Figure 6D*).

## The amplitude of head bending correlates with locomotory speed and efficiency

We showed that the extrasynaptic neurotransmission of the RME GABAergic neurons modulated the amplitude of head bending (*Figure 6D*). We then asked what role the amplitude of head bending played in forward locomotion. To address this question, we analyzed the relationship between head bending amplitude and the speed and efficiency of forward movement. In the case of slender organisms undulating in Newtonian fluids, the classical resistance force theory (*Gray and Hancock, 1955*) predicts quantitatively that the propulsion efficiency, which is defined as the ratio of the actual speed ($V_a$) to the wave speed of body undulation ($V_w$) in the reference frame of a worm, grows with the angle of attack ($\theta_a$), which is defined as the mean angle of a body segment with respect to the direction of forward movement (*Fang-Yen et al., 2010*). In the small angle limit, the relationship is given by the red curve and the equation in *Figure 7A* (*Gray and Hancock, 1955*). Using wild-type animals we measured both variables during bouts of ($\sim$10–15 s) forward locomotion (The data points in *Figure 7A*), and found that experimental data closely fit the theoretical curve (*Figure 7A*). We measured the amplitude of head bending and found that angle of attack was also positively correlated with the amplitude of head bending (*Figure 7B*). Therefore, increasing head bending correlates with greater locomotion efficiency and faster actual speed of movement (*Figure 7C*). However, when the head bending amplitude continued to increase, the wave speed of body undulation decreased

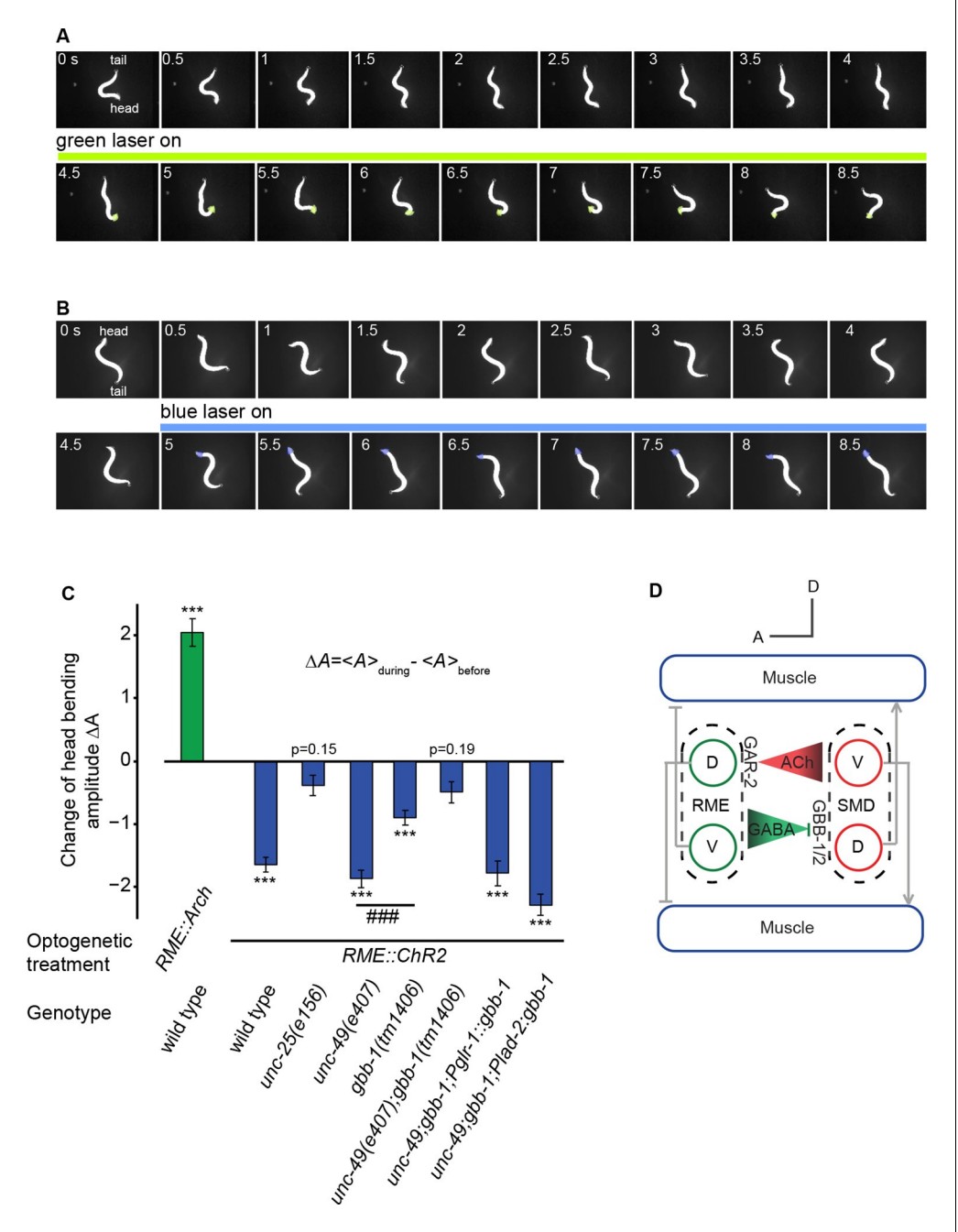

**Figure 6.** The activity of the RME GABAergic neurons is causally linked with head bending amplitude. (**A**) Video images of locomotory behavior before and during green light illumination in a worm that expresses Arch in RME neurons. The illuminated head region is highlighted in green. (**B**) Video images of locomotory behavior before and during blue light illumination in a worm that expresses ChR2 in RME neurons. The illuminated head region is highlighted in blue. (**C**) Quantification of optogenetic effects on head bending amplitude in wild type, mutants and transgenic animals that express a wild-type *gbb-1* cDNA under the *glr-1* or *lad-2* promoters in *unc-49;gbb-1* background. Sample size: *RME::Arch* in wild type (n = 25 trials, 15 animals), *RME::ChR2* in wild type (n = 98 trials, >20 animals), *RME::ChR2* in *unc-25(e156)* (n = 63 trials, 14 animals), *RME::ChR2* in *unc-49(e407)* (n = 85 trials, 17 animals), *RME::ChR2* in *gbb-1(tm1406)* (n = 81 trials, 11 animals), *RME::ChR2* in *unc-49(e407); gbb-1(tm1406)* (n = 83 trials, 6 animals), *RME::ChR2* in *unc-49(e407); gbb-1(tm1406); Pglr-1::gbb-1* (n = 51 trials, 7 animals), *RME::ChR2* in *unc-49(e407); gbb-1(tm1406); Plad-2::gbb-1* (n = 90 trials, 9 animals). For each trial, we calculate the change of the head bending amplitude during and before optogenetic manipulation of RME with sign test for zero median, ***p<0.0001; *unc-49(e407)* and *gbb-1(tm1406)* are compared with Mann-Whitney U test, ###p<0.0001; Mean ±

*Figure 6 continued on next page*

*Figure 6 continued*
SEM; while similar effects were usually observed in more than one transgenic lines, the effect of each transgene is reported with the results from one transgenic line. (D) A schematic model for the mechanisms underlying RME GABAergic neuromodulatory function.

(*Figure 7C*), possibly due to limiting biomechanical factors, such as speed of muscle contraction. As a result, within the physiological range of head bending amplitude, the speed of forward movement approached its maximum when the angle of attack was at around 45 degree (*Figure 7C*). These findings suggest that an optimal range of head bending amplitude might correlate with efficient forward movement. Consistently, we found that either optogenetically inhibiting or activating RME, which caused increased or decreased head bending amplitude (*Figure 6C*), significantly reduced the actual speed of locomotion (*Figure 7D*) and the propulsion efficiency (*Figure 7—figure supplement 1*). Together, our results reveal that the RME GABAergic modulatory neurotransmission plays an important role in setting the amplitude of head bending near the optimal value for forward movement.

## Discussion

To execute purposeful movement, a nervous system provides precise and sometimes optimal control of locomotory patterns. Neuromodulation, mediated by a large number of neurotransmitters and neuropeptides, plays a critical role in this regulation. Here, we characterized a modulatory role of the GABAergic motor neurons RME in regulating the undulatory head bending during forward movement. We identified the extrasynaptic mechanism through which RME inhibited the cholinergic excitatory motor neurons SMD via the GABA_B receptor GBB-1/GBB-2 to limit the amplitude of head bending. Interestingly, we also found that the activity of RME depended on the excitatory cholinergic inputs from SMD that also innervated the anterior muscles, providing a circuit mechanism whereby the activity of the GABAergic RME neurons temporally matches the activity of SMD and head bending undulation. We demonstrated that activating or inhibiting the activity of RME led to decreased or increased head bending amplitude during undulatory movement, causally linking the modulatory role of RME GABAergic signal with the amplitude of head undulation in moving animals.

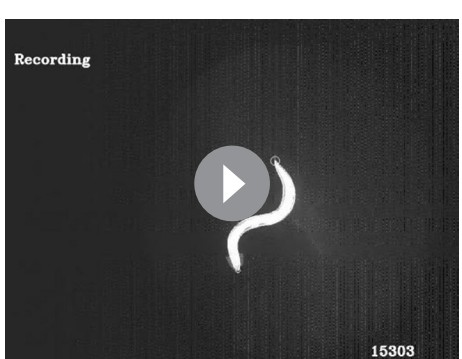

**Video 8.** Inhibiting RME activity with a green laser in a transgenic animal that expresses Arch in RME increases head bending amplitude. Laser illumination starts when a 'DLP ON' signal appears in upper-left corner and stops when the 'DLP ON' signal disappears. The worm is stimulated for one cycle of illumination, 5 s–34 s. The animal's head is pointing down at the beginning of the movie. The head region illuminated with green laser is highlighted with a shade. The tail is highlighted with a circle. The animal moves in 25%-30% (w/v) dextran sandwiched between two glass slides (Materials and methods).

While the undulatory forward movement in *C. elegans* generates a recurring locomotory pattern, the underlying rhythmogenic circuit or neurons have not been identified. Reciprocal inhibition, a common type of connectivity for rhythmogenic circuits that usually signal through glycinergic neurotransmission (*Alford and Williams, 1987*; *Brown, 1914*; *Buchanan, 1982*; *Cohen and Harris-Warrick, 1984*; *Dale, 1985*; *Friesen, 1994*; *Sharp et al., 1996*; *Soffe, 1987*), has not been identified in the *C. elegans* nervous system (*White et al., 1986*). Killing RME does not eliminate the undulatory forward movement, indicating that RME are not required to generate forward movement and revealing a modulatory role of the RME GABAergic neurotransmission in mediating the pattern of forward locomotion. It has been shown that GABA modulates locomotory circuits either extrinsically or intrinsically as part of the motor circuit (*Cazalets et al., 1998*; *Reith and Sillar, 1999*; *Tegner et al., 1993*; *Ziskind-Conhaim, 2013*; *Fidelin et al, 2015*). Here, we characterized a signaling mechanism whereby the RME GABAergic neurotransmission modulated

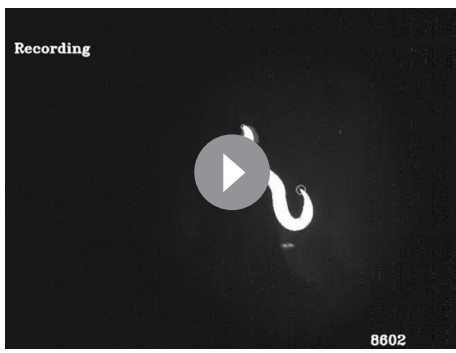

**Video 9.** Activating RME activity with a blue laser in a transgenic animal that expresses ChR2 in RME decreases head bending amplitude. The RIS neuron is ablated in the animal. Laser illumination starts when a 'DLP ON' signal appears in the upper-left corner and stops when the 'DLP ON' signal disappears. The worm is stimulated for 2 cycles of illumination, 4 s–15 s and 25 s–36 s. The head is pointing towards the upper-left corner at the beginning of the movie. The head region illuminated with blue laser is highlighted with a shade. The tail is highlighted with a circle. The animal moves in 25%–30% (w/v) dextran sandwiched between two glass slides (Materials and methods).

the pattern of head undulation during forward movements. We also demonstrated the causal role of the RME GABAergic neurotransmission in setting the amplitude of head bending in moving animals and showed that an optimal range of head bending amplitude correlated with the speed and efficiency of forward movement. We propose that the inhibitory feedback between SMD and RME provides a gain control mechanism to dynamically regulate head bending. When SMD activity is high and the head bending amplitude is large, RME inhibition is strong; when SMD activity is low and the head bending amplitude is small, RME inhibition is weak. This regulatory mechanism together with the phase-lock of RME activity to the head bending allows RME to continuously adjust its strength of inhibition based on the strength of SMD activity to set the head bending amplitude within an optimal range. Gain control mechanisms are commonly observed in motor systems (*Johnson and Heckman, 2014*). Here, we characterize the extrasynaptic neurotransmission of GABA, a common transmitter, that regulates a motor gain control through a conserved metabotropic GABA$_B$ receptor.

Both GABA$_A$ and GABA$_B$ receptors are implicated in modulating locomotory networks. Pharmacologically manipulating the activity of GABA$_A$ versus GABA$_B$ often generates different effects on locomotory circuits (*Cazalets et al., 1998*; *Reith and Sillar, 1999*; *Swensen et al., 2000*; *Tegner et al., 1993*; *Ziskind-Conhaim, 2013*). While GABA$_A$ receptors act as chloride channels; metabotropic GABA$_B$ receptors signal through G-protein pathways (*Olsen, 1991*). Here, we characterize the extrasynaptic GABAergic neurotransmission through which the *C. elegans* GABA$_B$ receptor modulates locomotion. *C. elegans* has two subunits of GABA$_B$ that are encoded by *gbb-1* and *gbb-2* and act as heterodimers (*Dittman and Kaplan, 2008*). We showed that the GABA$_B$ receptor subunit GBB-1 mediated the causal behavioral effect of the RME GABAergic neurotransmission on head bending undulation. In contrast, we found that mutating *unc-49*, which encoded a *C. elegans* homolog of GABA$_A$ (*Bamber et al., 1999*), in the wild-type background did not significantly reduce the effect of activating RME with ChR2 on the amplitude of head bending (*Figure 6*). It was previously shown that the GABA$_A$ receptor UNC-49 was expressed in the body wall muscles (*Bamber et al., 1999*) and mutating *unc-49* eliminated GABA-induced hyperpolarization in the body muscles (*Richmond and Jorgensen, 1999*), resulting in a 'shrinker' phenotype that phenocopied the locomotory defect of killing D-type GABAergic ventral nerve cord motor neurons, but not the defect of killing RME (*McIntire et al., 1993a*). Together, these findings indicate that both the identity of the GABAergic neurons and the target cells that express either GABA$_A$ or GABA$_B$ receptors contribute to the behavioral effects of GABAergic neurotransmission. We speculate that the slow kinetics of the metabotropic GABA$_B$ receptor and the downstream signaling match the temporal dynamics of head undulation, which operates at around 1 Hz (*Wen et al., 2012*).

Interestingly, the cholinergic neurotransmission from SMD that regulates the undulatory activity of RME is also mediated by a G-protein coupled receptor, GAR-2 (*Figure 4*). The protein sequence of GAR-2 is similar to the mammalian M2/M4 muscarinic acetylcholine receptors, which are coupled with G$_{i/o}$, and the mammalian M1/M3/M5 muscarinic acetylcholine receptors, which are coupled with G$_q$ (*Lanzafame et al., 2003*; *Lee et al., 2000*). Previously, it was shown that the function of *gar-2* in the cholinergic ventral nerve cord motor neurons genetically interacted with *goa-1*, which encoded the *C. elegans* G$_o$ (*Dittman and Kaplan, 2008*). Here, we showed that GAR-2 mediated the undulatory activity of RME in response to the cholinergic neurotransmission of SMD. We speculate that

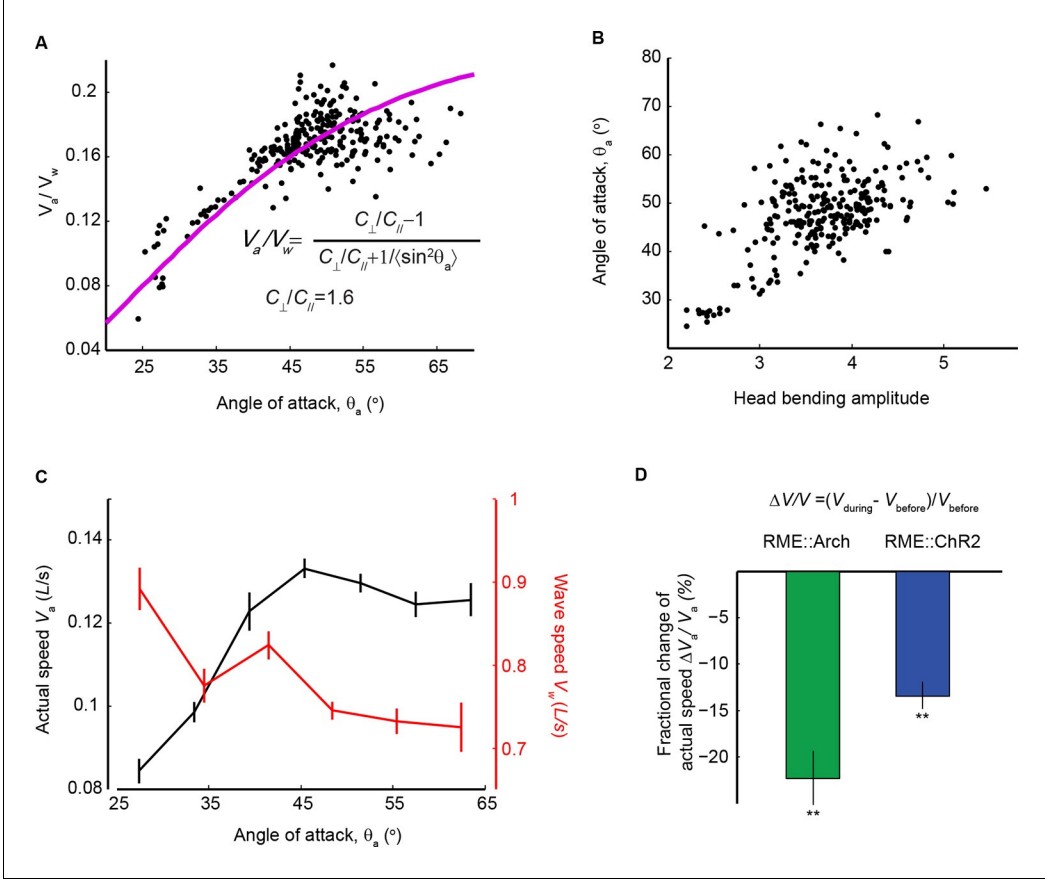

**Figure 7.** Head bending amplitude correlates with speed and efficiency of forward locomotion. (A) The angle of attack, measured as the average angle of a body segment with respect to the direction of forward movement, is plotted with the propulsion efficiency, defined as the ratio of the actual speed $V_a$ to the propagation speed of undulation in the reference frame of a worm $V_w$. Magenta line is the theoretical fit according to the equation in the figure panel. In this equation, $C_\perp$ is the drag coefficient perpendicular to the worm body and $C_{//}$ is the drag coefficient longitudinal to the worm body. $C_\perp/C_{//}$ = 1.6, which is the best fitting parameter, agrees well with theoretical calculation in the small angle limit (*Gray and Hancock, 1955*). (B) The angle of attack increases with head bending amplitude ($R$ = 0.57, p<10$^{-23}$, Pearson's correlation). In A and B, each dot represents one measurement from a bout (~10–15 s) of forward locomotion of wild-type animals in 25% dextran (w/v). (C) By binning the velocity data in (A) and (B), we replot the actual speed ($V_a$, black) and propagation wave speed of forward locomotion ($V_w$, red) with the angle of attack. Propagation wave speed decreases with the angle of attack ($R$ = −0.32, p<10$^{-6}$, Pearson's correlation). At around 45 degree angle, the actual speed of locomotion is maximized. For A-C, $n$ > 100 bouts of forward movements from 10 worms, Mean ± SEM. (D) Optogenetic inhibition of RME neurons that express Arch ($n$ = 11 trials, 5 animals) or optogenetic activation of RME neurons that express channelrhodopsin ChR2 ($n$ = 34 trials, 5 animals) significantly reduces the actual speed of forward locomotion. For each trial, we calculate the change of actual speed of locomotion during and before optogenetic manipulation of RME, **p<0.01, Wilcoxon signed rank test for zero median, Mean ± SEM. *Figure 7—figure supplement 1* shows the effects of optogenetic manipulation of RME activity on propulsion efficiency.

The following figure supplement is available for figure 7:

**Figure supplement 1.** Optogenetically inhibiting RME neurons that express Arch (n = 11 trials, 5 animals) or optogenetically activating RME neurons that express channelrhodopsin ChR2 (n = 34 trials, 5 animals) significantly reduces propulsion efficiency of forward locomotion.

GAR-2 might be coupled with an excitatory $G_\alpha$ subunit or might inhibit an inhibitory signaling pathway, resulting in the activation of RME. Alternatively, GAR-2 may regulate the head-bending

correlated temporal dynamics of RME activity and loss of GAR-2 would result in the reduced correlation of RME activity and head undulation, altering the modulatory effect of RME on head bending.

There are 26 GABAergic neurons in *C. elegans,* including 4 RME, 19 D-type ventral nerve cord motor neurons and interneurons AVL, DVB and RIS (*McIntire et al., 1993b*). While our study addresses the modulatory role of RME in regulating the amplitude of head undulation and how changing head bending amplitude correlates with propulsion efficiency, it does not exclude other mechanisms whereby GABA regulates locomotory patterns. For example, it was previously shown that the D-type GABAergic ventral nerve cord motor neurons directly synapsed onto the cholinergic ventral nerve cord motor neurons and inhibited these excitatory motor neurons through the GBB-1/GBB-2 receptor to mediate the speed of forward movement (*Dittman and Kaplan, 2008*). Our results together with these previous findings highlight the importance of the underlying circuit connectivity in shaping the behavioral effects of neuromodulation.

Among the RME neurons, there are RMED (dorsal), RMEV (ventral), RMEL (left) and RMER (right). Here, we showed that it was the GABAergic neurotransmission from RMED and RMEV, but not RMEL or RMER, that modulated head undulation. What accounts for the functional difference between the dorsal-ventral pair and the left-right pair of RME neurons? Previous studies in the stomatogastric nervous system suggested that difference in the diffusion patterns of the neuropeptide proctolin that was released from different modulatory neurons contributed to different modulatory effects of proctolin (*Wood et al., 2002*). There is only one synapse between RMED/V neurons and four SMD motor neurons, two SMDD (dorsal) and two SMDV (ventral), consistent with the involvement of extrasynaptic neurotransmission. The somata of RMED and RMEV are located next to the somata of SMDD and SMDV on the dorsal and ventral sides; while the somata of RMEL and REMR are mainly on the lateral sides, away from SMD (*White et al., 1986*). Thus, it is conceivable that the anatomical features of these motor neurons allow the GABA signal released from RMED/V to regulate SMD more efficiently than the GABA released from RMEL/R, highlighting the importance of the spatial property of neuromodulators in their modulatory effects on neural circuits and behavior. It is also possible that GABA in RMED/V versus RMEL/R is co-released with different neurotransmitters or peptides, a mechanism which also partially accounts for different modulatory effects of proctolin released from different modulatory neurons in the stomatogastric nervous system (*Nusbaum et al., 2001*). A complete profile of the neurotransmitters and peptides expressed in the four RME neurons is not available yet; therefore, the potential difference in the co-release of neurotransmitters from different RME neurons remains to be further investigated.

Our study reveals that the SMD and RME motor neurons signal to each other through extrasynaptic neurotransmission and metabotropic receptors to modulate a locomotory pattern. This neuromodulatory signal forms a functional circuit that cannot be predicted by using the anatomical connections described by the wiring diagram. Therefore, our results highlight the critical role of neuromodulation in shaping the information flow of the nervous system. All nervous systems express a large number of neuromodulators that can substantially regulate the functional organization of neural circuits. Our study points to the importance of functional studies in combination with wiring diagrams in order to better understand how the nervous systems operate.

# Materials and methods

## Strains

*C. elegans* strains were raised under the standard conditions at 20°C (*Brenner, 1974*). Strains used in this study include: N2 Bristol (wild type), CB156 *unc-25(e156)*III, CB307 *unc-47(e307)*III, EG5025 *oxIs351[Punc-47::ChR2::mCherry::unc-54 3'UTR; lin-15(+)*; LITMUS 38i]X, FX01406 *gbb-1(tm1406)*X, KP3447 *nuEx1066[Pgbb-1::gfp; Pttx-3::dsRed2]*, KP6566 *gbb-1(tm1406)*; *gbb-2(tm1165)*IV, ST2351 *ncEx2351[Punc-47::Arch::eGFP; Pmyo-2::mCherry]*, ZC1148 *yxIs1[Pglr-1::GCaMP3.3; Punc-122::gfp]* V, ZC361 *lin-15B(n765)*X; *kyIs30[Pglr-1::gfp; lin-15(+)]*X (CX2610 crossed with wild type), ZM6665 *hpIs268[Punc-25::GCaMP3::UrSL::wCherry]*, ZC1553 *yxEx750[Pglr-1::TeTx::mCherry; Punc-122::gfp]*, ZC2181 *unc-13(e51)*I; *hpIs268*, ZC2204 *cha-1(p1152)*IV; *hpIs268*, ZC2211 *cha-1(p1152)*IV; *hpIs268*; *yxEx809*[cosmid ZC416, *Punc-122:gfp*], ZC2234 *hpIs268; yxEx1176[Podr-2(18)::TeTx::mCherry; Punc-122::dsRed]*, ZC2258 *yxEx1197[Pklp-6::TeTx::mCherry; Punc-122::dsRed]*, ZC2264 *hpIs268; yxEx1190 [Pklp-6::gfp; Punc-122::dsRed]*, ZC2273 *hpIs268; yxEx1197*, ZC2298 *nuEx1066; yxEx1154[Pglr-1:*

mcherry; Punc-122::gfp], ZC2299 hpls268; yxEx1176; yxEx1197, ZC2327 cha-1(p1152)IV; hpls268; yxEx1211[Podr-2(18)::cha-1::mCherry; Punc-122::dsRed], ZC2329 cha-1(p1152)IV; hpls268; yxEx1213 [Pklp-6::cha-1::mCherry; Punc-122::dsRed], ZC2330 cha-1(p1152)IV; hpls268; yxEx1214[Pglr-1::cha-1::mCherry; Punc-122::dsRed], ZC2332 yxls1/hpls268, ZC2336 hpls268; yxEx750, ZC2363 hpls268; yxEx778[Plad-2::TeTx::mCherry; Punc-122::gfp], ZC2374 gbb-1(tm1406)X; yxEx1234[Pglr-1::gbb-1; Punc-122::dsRed], ZC2375 cha-1(p1152)IV; hpls268; yxEx1235[Plad-2::cha-1::mCherry; Punc-122::dsRed], ZC2383 gbb-1(tm1406)X; yxEx1243[Pgbb-1::gbb-1; Punc-122::dsRed], ZC2404 gbb-1(tm1406)X; yxEx1254[Plad-2::gbb-1; Punc-122::dsRed], ZC2422 unc-49(e407)III; oxls351, ZC2423 gbb-1(tm1406)X; oxls351, ZC2430 yxls30[Punc-47::Arch::eGFP; Pmyo-2::mCherry], ZC2432 unc-49 (e407)III; gbb-1(tm1406)X; oxls351, ZC2433 unc-25(e156)III; oxls351, ZC2446 unc-49(e407)III; gbb-1 (tm1406)X; oxls351, ZC2447 unc-49(e407)III; gbb-1(tm1406)X; oxls351; yxEx1234, ZC2618 gar-2(ok520)IV, ZC2622 hpls268; gar-2(ok520)IV; nuEx1075[Pgar-2::venus::gar-2cdna]. For behavioral analysis, unc-25(e156)III and gbb-1(tm1406)X were at least 2× outcrossed with wild type. The extra-chromosomal array ncEx2351[Punc-47::Arch::eGFP; Pmyo-2::mCherry] was integrated by a UV cross-linker (Stratalinker 2400, energy setting 300) and 6× outcrossed with wild type to generate yxls30.

## Molecular biology

Molecular cloning in this study was performed using the Gateway system (Invitrogen) unless otherwise specified. The 1.9 kb cha-1 cDNA was cut by NotI and AgeI enzyme from Podr-2::cha-1::gfp (a gift from J. Lee) and ligated upstream of mCherry to produce pPD95.77-cha-1::mCherry, and a Gateway recombination cassette (rfB) was ligated upstream of the cha-1 cDNA to produce pPD95.77-rfB-cha-1::mCherry. The 2565 bp gbb-1 cDNA was cut by NheI and KpnI from a pBlue-Script vector (KP#1026, a gift from J. Dittman) and cloned into the pPD49.26 vector to produce pPD49.26-rfB-gbb-1. The destination vectors pPD95.77-rfB-gfp, pPD95.77-rfB-mCherry and pSM-rfB-TeTx::mCherry were generated previously in the lab (*Hendricks et al. 2012*). A 2.4 kb genomic fragment upstream of the odr-2 gene (forward primer 5' AGT TCA CCA AGC TCT TCT CGT TTA TTC, reverse primer 5' CCA TCA GCC AAA TGT AGG CTC GGT TC), 1.5 kb upstream of the klp-6 gene (forward primer 5' CAC CAA AAA ATT CAT TAA, reverse primer 5' TAT TCT GAA AAG TTC AAC TAA TA), and 3 kb upstream of the gbb-1 gene (forward primer 5' CGT CGT TCT CAT TGT ATG CCG TTT AAC, reverse primer 5' CGG AAA CGT GCC ACC GAT GTG AAG) were amplified by polymerase chain reaction (PCR) and ligated to the pCR8 backbone to produce the Gateway entry vectors. The pCR8-Pglr-1 and pCR8-Plad-2 entry clones were generated previously in the lab (*Hendricks et al., 2012*). LR recombination reactions were performed using Gateway LR Clonase enzyme kits according to the protocol provided (Invitrogen), generating the expression clones Podr-2(18)::cha-1::mCherry, Pklp-6::cha-1::mCherry, Pglr-1::cha-1::mCherry, Plad-2::cha-1::mCherry, Podr-2(18)::TeTx::mCherry, Pklp-6::TeTx::mCherry, Pklp-6::gfp, Podr-2(18)::gfp, Pglr-1::mCherry, Pgbb-1::gbb-1, Pglr-1::gbb-1, and Plad-2::gbb-1. To generate transgenic lines, Pgbb-1::gbb-1 was injected at 1 ng/μl. Pglr-1::gbb-1 and Plad-2::gbb-1 were injected at 10 ng/μl and the other plasmids were injected at 25 ng/μl. Microinjection was performed as described (*Mello et al., 1991*) with either Punc-122::gfp or Punc-122::dsRed as a co-injection marker for all the transgenic lines.

## Laser ablation

Laser ablation was performed as previously described with slight modifications (*Bargmann and Avery, 1995*; *Fang-Yen et al., 2012*). UV laser pulses (Micropoint, Andor) were focused onto neurons of interest through a 60× water immersion objective on an upright Nikon microscope (Eclipse LV150). Target neurons were identified by using anatomical characteristics in combination with fluorescence reporters that were expressed with cell-specific promoters. Laser ablation was performed on L1-L2 larvae. Animals that underwent laser ablation or mock procedures were recovered and raised at 20°C under standard conditions for 2 days till they reached adulthood. After behavioral assays, each ablated animal was recovered and the expression of the fluorescent reporter was examined under a 40× objective on a Nikon TE2000-U microscope. Animals with remaining fluorescent signals in target neurons were excluded from analysis. The n numbers in the relevant figure legends denote the numbers of different animals for the experiments and represent independent biological replication.

## Calcium imaging

Calcium imaging was performed in a microfluidic device as previously described (*Chronis et al., 2007*; *Hendricks et al., 2012*) with modifications. A SU-8 master was used to cast Polydimethylsiloxane (PDMS) (Dow Corning Sylgard 184, Ellsworth Adhesives, Germantown, WI) chips and inlet holes were drilled manually. Chips were bonded to glass coverslips with a handheld corona treater and connected to a perfusion system (Automate Scientific Valvebank, Berkeley, CA). Fluorescence time-lapse imaging (100–200 ms exposure, 5 Hz) was performed on a Nikon Eclipse TE2000-U inverted microscope with a 40× oil immersion objective and a Photometrics CoolSnap EZ camera. To image neurons on different focal planes, an image stack was collected on a confocal spinning disk microscope. Animals were washed by NGM buffer (1 mM $CaCl_2$, 1 mM $MgSO_4$, 25 mM $KPO_4$ pH6.0) briefly before being loaded in the worm channel, and were presented with slow streams of NGM buffer throughout the recording of up to 5 min. All image analysis was performed with ImageJ (NIH) unless otherwise specified. Frames were aligned using the StackReg plugin where necessary. For RME imaging, total fluorescence intensity was measured by subtracting background from the region of interest (ROI) corresponding to the cell body of RMED or RMEV neuron during head bending. For cross-correlation analysis, the fluorescence intensity data were normalized for each individual to a linear scale by the formula $(F - F_{min})/(F_{max} - F_{min})$ (*Hendricks et al., 2012*). For simultaneous imaging of SMD and RME, image stacks were composed and processed by a customized program (uploaded to https://github.com/Wenlab/worm-imaging-analysis) written in Matlab (Mathworks, Natick). Imaging on immobilized animals were performed on a 10% agarose pad with 0.3–0.5 µl of 0.1 µm diameter polystyrene microspheres (Polysciences 00876–15) (*Kim et al., 2013*). The n numbers in the relevant figure legends denote the numbers of different animals for the experiments and represent independent biological replication.

## Head movement analysis

Head bending of animals during calcium imaging in the microfluidic chip was quantified with ImageJ (NIH, MD) essentially as described (*Hendricks et al., 2012*). Images were binarized with all background pixels converted to 0 and all pixels representing the animal converted to 1. An ROI comprising the moving part of the animal's head was selected and fit into an ellipse in ImageJ, where the orientation of the ellipse was measured for each frame. The difference in orientation between individual frames and the reference frame (minimum head bending) was calculated, generating either positive or negative values that represented head deflection along the ventral-dorsal axis. For cross-correlation analysis, the head bending was then normalized to the maximum deflection, generating an index between −1 and 1. Ventral bending was defined as positive. The n numbers in the relevant figure legends denote the numbers of different animals for the experiments and represent independent biological replication.

## Cross-correlation analysis

Cross-correlation between calcium signals and head bending was analyzed in JMP10 software (SAS) as described (*Hendricks et al., 2012*) with modifications. Time series data of fluorescence intensity and head position were normalized within each individual and taken for cross-correlation with a time lag of 20 s (100 frames). Head position was used as input. Comparison between peak correlations of different genotypes was performed as described (*Hendricks et al., 2012*). Briefly, the peak of the mean wild-type control correlation was identified at time $T_p$ and the maximum correlation, positive or negative, was taken from strains of interest in a 1 s time window centered on $T_p$. Unless otherwise specified, cross-correlation between RMEV calcium activity and head bending amplitude was analyzed, mutants were compared with wild-type controls and transgenic animals were compared with non-transgenic siblings by Student's t-test or Analysis of variance (ANOVA). The n numbers for cross-correlation and peak correlation in the relevant figure legends denote the numbers of different animals for the experiments and represent independent biological replication.

## Tracking and analysis of head bending amplitude

Tracking of freely-moving animals were performed on a custom-built single-worm tracker. Well-fed adult hermaphrodites were washed briefly in NGM buffer and placed onto a 9 cm NGM agar plate without food. Animals were allowed to move freely on the plate for 2 min before being recorded for

at least 1 min. The plate was placed on a moving stage under a DMK 21AU04 monochrome scan camera (Imaging Source, 30 Hz) and was illuminated by infrared light from a LED ring. Images were captured with a customized program (*Source code 1*) written in LabVIEW8.5 (National Instrument) and saved every 6 frames. The video recorded was processed by a customized program (*Source code 2*) written in Matlab (Mathworks, Natick) (*Fang-Yen et al., 2010*; *Wen et al., 2012*). Frames in which animals were making reversals or turns were excluded and at least 150 frames (30 s) per trial were used for analysis. In brief, the head and tail of an animal were identified as the points of maximum convex curvature along the animal's boundary and confirmed manually. The centerline of the animal's body was extracted and smoothened, and divided into 100 segments. The first 18 segments, or the first 18% of the whole body-length from the nose tip, were counted as the head region, and the head curvature of an animal was computed as the average head curvature of the 18 segments. The head bending amplitude was computed as the standard deviation of the head curvature during the period of measurement (*Figure 1B*). The n numbers in the relevant figure legends denote the numbers of different animals for the experiments and represent independent biological replication.

## Fluorescence microscopy

Fluorescent images were collected with a Nikon Eclipse TE2000-U microscope with a 40× oil immersion objective. Images were processed with Image J (NIH).

## Optogenetics

Optogenetic stimulation of RME neurons in freely moving animals was performed on a custom-built structured illumination system (*Leifer et al., 2011*), which consists of an Nikon TE2000 inverted microscope, a high speed CCD camera, blue and green solid state lasers, and a digital micromirror device, all of which are controlled by the MindControl software (*Leifer et al., 2011*; the software is publisehd and available at http://www.nature.com/nmeth/journal/v8/n2/full/nmeth.1554.html). Animals used in all the optogenetics experiments were raised in the dark at 20°C on NGM plates with *E. coli* OP50 and all-*trans* retinal. The OP50-retinal plates were prepared 1–2 days in advance by seeding a 6 cm NGM-agar plate with 250 µl of OP50 culture and 1 µl of 100 mM retinal dissolved in ethanol. Adult animals were washed briefly in NGM buffer and transferred into a layer of 25% or 30% dextran (w/v) sandwiched between two glass slides separated by a 75 µm spacer. The animal was slightly compressed to reduce head movement in the z direction. During the assay, an individual animal was imaged under infrared light with dark field illumination, and the first 10% of the body-length from nose tip was targeted by a laser beam reflected by the digital micromirror device. For RME inhibition by Arch, the green laser (532 nm) was switched on for seconds in each trial. For RME activation by ChR2, the blue laser (473 nm) was switched on for seconds in each trial, and the RIS interneuron was removed by laser surgery in larvae at the L1-L2 stage. In the relevant figure legends, the n numbers denote the numbers of different trials on indicated numbers of different animals for the experiments, and represent both independent biological replication and technical replication.

## Acknowledgements

We thank the *Caenorhabditis* Genetics Center (funded by NIH Office of Research Infrastructure Programs P40 OD010440) and the *C. elegans* Gene Knockout Consortium for strains, the Wellcome Trust Sanger Institute for cosmids, Drs. J Dittman, N Ji, J Kaplan, J Lee, A. Leifer, L Looger, and M Zhen for sharing reagents and protocols, Drs. ER Soucy and J Greenwood for building the tracking system and programs, and B Han for helping with figure illustration. We also thank the Zhang lab members for discussion. Q Wen is supported by National Nature Science Foundation of China (NSFC-31471051). T Kawano was supported by grants from CIHR and NSERC (MOP-123250 and MOP-93619 to M Zhen). The research in the Samuel laboratory was supported by National Institutes of Health (8DP1GM105383 and 1P01GM103770) and National Science Foundation (PHY-0957185). The research in the Zhang laboratory was supported by John Merck Fund and National Institutes of Health (NIDCD DC009852).

# Additional information

## Funding

| Funder | Grant reference number | Author |
| --- | --- | --- |
| National Institute on Deafness and Other Communication Disorders | NIDCD DC009852 | Yu Shen<br>He Liu<br>Yuqi Qin<br>Gareth Harris<br>Yun Zhang |
| National Institute of General Medical Sciences | 8DP1GM105383 | Quan Wen<br>Aravinthan DT Samuel |
| National Natural Science Foundation of China | NSFC-31471051 | Quan Wen<br>Min Wu<br>Tianqi Xu |
| National Institute of General Medical Sciences | 1P01GM103770 | Quan Wen<br>Aravinthan DT Samuel |
| Canadian Institutes of Health Research | | Taizo Kawano |
| Natural Sciences and Engineering Research Council of Canada | | Taizo Kawano |
| National Science Foundation | PHY-0957185 | Aravinthan DT Samuel |
| John Merck Fund | | Yun Zhang |

The funders had no role in study design, data collection and interpretation, or the decision to submit the work for publication.

## Author contributions

YS, QW, HL, Conception and design, Acquisition of data, Analysis and interpretation of data, Drafting or revising the article, Contributed unpublished essential data or reagents; CZ, YQ, Conception and design, Acquisition of data, Analysis and interpretation of data, Contributed unpublished essential data or reagents; GH, Conception and design, Acquisition of data, Analysis and interpretation of data, Drafting or revising the article; TK, Taizo Kawano shared unpublished reagents that generated the results included in 4 out of the 7 main figures of the paper; MW, TX, Acquisition of data, Analysis and interpretation of data, Drafting or revising the article, Contributed unpublished essential data or reagents; ADTS, YZ, Conception and design, Analysis and interpretation of data, Drafting or revising the article

## Author ORCIDs

Yun Zhang, http://orcid.org/0000-0002-7631-858X

# Additional files

## Supplementary files

• Source code 1. auto tack 23.vi. The LabVIEW program used to record movement of *C. elegans*.

• Source code 2. YS_test_analyze_agarose_free_moving.m. The Matlab code used to quantify head bending amplitude in moving animals.

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
