## [Decision Letter]

Thank you for submitting your article "An extrasynaptic GABAergic signal modulates a locomotory gait" for consideration by *eLife*. Your article has been reviewed by two peer reviewers, and the evaluation has been overseen by a Ronald L. Calabrese as the Reviewing Editor and Eve Marder as the Senior Editor.

The reviewers have discussed the reviews with one another and the Reviewing Editor has drafted this decision to help you prepare a revised submission.

Summary:

This is a very thorough, methodical determination of connectivity between two neurons involved in locomotion in *C. elegans*, SMD and RME. The study demonstrates that SMD uses cholinergic signaling to activate RME which in turn inhibits SMD through primarily GABAB receptor-dependent signaling. Further the role of either increasing or decreasing activity in these cells, including the consequences of ablating neurons or disrupting chemical messenger pathways is nicely demonstrated in behaving animals using optogenetic techniques. Most conclusions of the study are very well supported by the data. The use of every imaginable advantage of the worm system to manipulate and assess neuronal circuit activity and behavior, all in live animals, is clearly the major strength of this paper. The paper is yet another demonstration of the limits of connectomics since extrasynaptic signals such as those described here follow no anatomical rules.

The figures presented are detailed and convincing and supported by ample supplemental data (these could be better integrated into the main text). Writing is in general clear.

Essential revisions:

The reviewers were in agreement about the thoroughness of the work, but were concerned about two issues which must be addressed before this paper can be published in *eLife*.

1) The authors have to indicate for each figure panel in which they used transgenic lines whether these results were observed with more than one transgenic line, as is standard in the field and is required for publication.

2) Discussion is too inwardly focused on issues of concern to the *C. elegans* community and misses an opportunity to reach out to a larger audience. Two issues should be addressed directly:

A) Quoting from the review discussion: "…the aspect that I would like to see discussed is what their data means in terms of interpreting connectome data. People seem so obsessed with connectomes these days that they tend to forget that even a connectome as well mapped as that of the worm will not predict who talks to whom. That may not come as a surprise, but this manuscript is terrific in the way it shows this."

B) Quoting from the review discussion: "…enthusiasm and the broader interest of this manuscript would be increased by at least speculation about the functional implications of the circuitry…". To guide this revision see the review comments below:

"This small circuit involves a neuron which elicits head bending, activating another neuron which decreases head bending. If this is a static component of the system, it is a convoluted way to elicit the correct amount of head bending. The Discussion would benefit from some discussion of this circuit and potential utility of this arrangement. I would favor discussion of functional implications of the circuitry over discussion of details such as hypothesizing about roles of other individual neurons in the system (Ex. Discussion, fifth and sixth paragraphs)."

"Why isn't there just 1 cell that elicits the appropriate level of head bending? Why is the GABAergic modulation time-locked to the head bending; would a tonic inhibitory modulatory tone have different consequences?"

Other concerns:

1) In the subsection “Optogenetic analysis of the RME GABAergic modulatory signal“. If the dual mutant without GABAA or GABAB "completely abolishes" the effects of RME, but GABAB mutant only significantly decreases the effects, doesn't that suggest that there is a contribution of GABAA receptors? How does eliminating a GABAA receptor in muscles test its central role in modulation of the motor circuit?

2) In the subsection “The amplitude of head bending mediates locomotory speed and efficiency”. The data presented here are correlations between head bending and locomotor speed. How has the "effect" of head bending amplitude been determined? By altering RME activity, could there be other parallel effects on locomotion? Is the only effect of RME to alter head bending amplitude? It seems this sections needs to be re-written to indicate that correlations exist which are suggestive or it needs to be better explained why there is a definitive causal link between head bending amplitude and the speed of locomotion.

---

## [Author Response]

Essential revisions:

*The reviewers were in agreement about the thoroughness of the work, but were concerned about two issues which must be addressed before this paper can be published in eLife.*

*1) The authors have to indicate for each figure panel in which they used transgenic lines whether these results were observed with more than one transgenic line, as is standard in the field and is required for publication.*

While similar effects were observed in more than one transgenic lines, we reported the results from one line for each transgene in this manuscript. We have now clarified it as requested in the relevant figure legends.

The main approach we used in this manuscript to ensure robust results is two-fold. We support key conclusions with results that were generated from different transgenes and with results that were generated from experiments with different methods, specifically:

In Figure 2 and Figure 3, we used two transgenes, *Pglr-1::cha-1* and *Plad-2::cha-1*, that had overlapping expression of *cha-1* in the SMD neurons to demonstrate the role of the cholinergic transmission of SMD in regulating RME activity; in addition, to strengthen the critical role of the expression of *cha-1* in SMD in regulating RME activity, we also used laser ablation of SMD in the transgenic *cha-1* mutants that expressed *Pglr-1::cha-1*. To demonstrate the specific effects of the *Pglr-1::cha-1* and *Plad-2::cha-1* transgenes, we used two transgenes, *Pklp-6::cha-1* and *Podr-2(18)::cha-1*, as negative controls.

In Figure 4, to show the function of *gar-2* in head bending, we used two different transgenes, *Pgar-2::gar-2* and *Punc-25::gar-2*, both of which rescued the head bending phenotype in the *gar-2* mutant.

In Figure 4, to show the function of SMD in regulating RME activity, we used two transgenes, *Pglr-1::TeTx* and *Plad-2::TeTx*, to block the neurotransmitter release from SMD; we also used laser ablation to kill SMD. In addition, we used two different transgenes, *Pklp-6::TeTx* and *Podr-2(18)::TeTx*, as negative controls to show the specific effects of *Pglr-1::TeTx* and *Plad-2::TeTx*.

In Figure 5, to show the role of *gbb-1* expression in SMD in regulating head bending, we used two transgenes, *Pglr-1::gbb-1* and *Plad-2::gbb-1*, which had overlapping expression in SMD. We also used laser ablation of SMD to substantiate the role of SMD in regulating head bending.

In Figure 6 and Figure 7, to validate the causal role of RME in regulating head bending, we used two transgenic reagents, *RME::ChR2* and *RME::Arch,* to activate or inhibit RME, respectively.

*2) Discussion is too inwardly focused on issues of concern to the C. elegans community and misses an opportunity to reach out to a larger audience. Two issues should be addressed directly:*

*A) Quoting from the review discussion: "…the aspect that I would like to see discussed is what their data means in terms of interpreting connectome data. People seem so obsessed with connectomes these days that they tend to forget that even a connectome as well mapped as that of the worm will not predict who talks to whom. That may not come as a surprise, but this manuscript is terrific in the way it shows this."*

We appreciate the suggestion. We have now added discussion (Discussion section, last paragraph) on the limitation of connectomes in predicting functional connectivity.

B) Quoting from the review discussion: "…enthusiasm and the broader interest of this manuscript would be increased by at least speculation about the functional implications of the circuitry…". To guide this revision see the review comments below:

*"This small circuit involves a neuron which elicits head bending, activating another neuron which decreases head bending. If this is a static component of the system, it is a convoluted way to elicit the correct amount of head bending. The Discussion would benefit from some discussion of this circuit and potential utility of this arrangement. I would favor discussion of functional implications of the circuitry over discussion of details such as hypothesizing about roles of other individual neurons in the system (Ex. Discussion, fifth and sixth paragraphs)."*

*"Why isn't there just 1 cell that elicits the appropriate level of head bending? Why is the GABAergic modulation time-locked to the head bending; would a tonic inhibitory modulatory tone have different consequences?"*We appreciate the suggestion. We show that SMD facilitate head bending as well as activate RME, and RME subsequently inhibit SMD to limit head bending. We propose that this convoluted way of regulating head bending provides a gain control mechanism. When SMD activity is high and the head bending amplitude is large, RME inhibition is strong; when SMD activity is low and the head bending amplitude is small, RME inhibition is weak. This regulatory mechanism together with the phase-lock of RME activity to the head bending allows RME to dynamically adjust its strength of inhibition based on the strength of SMD activity to set the head bending amplitude within an optimal range. Gain control mechanisms are commonly observed in motor systems. In this paper, we have identified the role of a conserved B-type GABAergic receptor in mediating motor gain control. We propose that the circuit organization and the regulatory mechanisms revealed in our study would be of a general appeal. We have included the discussion in the second paragraph of the Discussion section.

Other concerns:

*1) In the subsection “Optogenetic analysis of the RME GABAergic modulatory signal“. If the dual mutant without GABAA or GABAB "completely abolishes" the effects of RME, but GABAB mutant only significantly decreases the effects, doesn't that suggest that there is a contribution of GABAA receptors? How does eliminating a GABAA receptor in muscles test its central role in modulation of the motor circuit?*

We agree that in the *gbb-1(tm1406)* mutant background there was a residual effect of ChR2 activation of RME, while the *unc-49(e407);gbb-1(tm1406)* double mutations completely abolished the effect, which suggests some potential role of the UNC-49 GABAA receptor in regulating head bending amplitude. However, on the other hand, the *unc-49(e407)* mutation alone did not seem to reduce the effect of activating RME with ChR2. We speculate that the role of UNC-49 GABAA receptor in regulating the behavioral effects of activating RME with ChR2 may only become detectable when the GBB-1 GABAB receptor is removed. However, the potential mechanism is currently unknown. We have further discussed these results in the last paragraph of the subsection “Optogenetic analysis of the RME GABAergic modulatory signal”.

*2) In the subsection “The amplitude of head bending mediates locomotory speed and efficiency”. The data presented here are correlations between head bending and locomotor speed. How has the "effect" of head bending amplitude been determined? By altering RME activity, could there be other parallel effects on locomotion? Is the only effect of RME to alter head bending amplitude? It seems this sections needs to be re-written to indicate that correlations exist which are suggestive or it needs to be better explained why there is a definitive causal link between head bending amplitude and the speed of locomotion.*

We appreciate the thoughts and comments. Our optogenetics experiments shown in Figure 6 in combination with molecular genetics, calcium imaging and behavioral analysis indicate that the activity of RME modulates head bending amplitude through an extrasynaptic signal of GABA. Meanwhile, we also found that when we changed RME activity with optogenetics, we could also change the speed and propulsion efficiency of forward movement (Figure 7 and Figure 7—figure supplement 1). In Newtonian fluid, the propulsion efficiency as a function of angle of attack can be derived from resistive force theory; this theoretical description fits well with our measurement on head bending and propulsion efficiency (Figure 7). Therefore, the amplitude of head bending strongly correlates with speed and propulsion efficiency, although we agree that we do not yet understand the mechanism underlying the correlation. We have now re-written the section to emphasize the correlation.